

# Exploring the daytime boundary layer evolution based on Doppler spectrum width from multiple coplanar wind lidars during CROSSINN

Nevio Babić[1,2], Bianca Adler[3], Alexander Gohm[4], Manuela Lehner[4], and Norbert Kalthoff[2]

[1]Croatia Control Ltd., Ulica Rudolfa Fizira 2, 10410, Velika Gorica, Croatia
[2]Institute for Meteorology and Climate Research, Karlsruhe Institute of Technology, Germany
[3]CIRES, University of Colorado Boulder, and NOAA/Physical Sciences Laboratory, Boulder, Colorado, USA
[4]Department of Atmospheric and Cryospheric Sciences, University of Innsbruck, Innsbruck, Austria

**Correspondence:** Nevio Babić (nevio.babic@crocontrol.hr)

**Abstract.** Over heterogeneous, mountainous terrain, the determination of spatial heterogeneity of any type of a turbulent layer has been known to pose substantial challenges in mountain meteorology. In addition to the combined effect in which buoyancy and shear contribute to the turbulence intensity of such layers, it is well known that mountains add an additional degree of complexity via non-local transport mechanisms, compared to flatter topography. It is therefore the aim of this study to determine the vertical depths of both daytime convectively and shear-driven boundary layers within a fairly wide and deep Alpine Valley during summertime. Specifically, three Doppler lidars deployed during the CROSSINN (Cross-valley flow in the Inn Valley investigated by dual-Doppler lidar measurements) campaign within a single week in August 2019 are used to this end, as they were deployed along a transect nearly perpendicular to the along-valley axis. To achieve this, a bottom-up exceedance threshold method based on turbulent Doppler spectrum width sampled by the three lidars has been developed and calibrated against a more traditional bulk-Richardson number approach applied to radiosonde profiles obtained above the valley floor. The method was found to adequately capture the depths of convective turbulent boundary layers at a 1-min temporal and 50-m spatial resolution across the valley, with the degree of ambiguity increasing once surface convection decayed and upvalley flows gained in intensity over the course of the afternoon and evening hours. Analysis of four Intensive Observation Period (IOP) events elucidated three regimes of the daytime mountain boundary layer in this section of the Inn Valley. Each of the three regimes has been analyzed as a function of surface sensible heat flux $H$, upper-level valley stability $\Gamma$, and upper-level subsidence $w_L$ estimated with the coplanar retrieval method. Finally, the positioning of the three Doppler lidars in a cross-valley configuration enabled one of the most highly spatially and temporally resolved observational convective boundary layer depth data sets during daytime and over complex terrain to date.

## 1 Introduction

On an hourly time scale or shorter, the effect of a homogeneous and horizontally flat (HHF) surface forcing defines the bulk characteristics of a layer adjacent to this surface, known as the atmospheric boundary layer (ABL; Stull, 1988). Although this definition is similar for complex, heterogeneous mountainous terrain, the temporal scales involved are longer by up to





a few hours, leading to the introduction of a mountain boundary layer (MoBL; Lehner and Rotach, 2018). In either case, a positive surface sensible heat flux during daytime will lead to pronounced convective overturning, leading to the convective

boundary layer (CBL). Here we focus on dry rather than on moist convection, as the latter often results in deep, precipitous convective cells (Kirshbaum et al., 2018), thereby perturbing the gentle balance of quiescent conditions necessary for fair-weather development of CBL and MoBL. While the assumption of vertical exchange being responsible for the majority of convectively-driven turbulence in the CBL is acceptable over HHF terrain, this cannot be expected to hold over complex terrain, where horizontal motions and associated effects become increasingly more important to consider (Rotach et al., 2015, 2022;

Lehner and Rotach, 2018). In addition, a multitude of non-turbulent transport mechanisms are made possible by the presence of mountain chains (Whiteman, 2000; Zardi and Whiteman, 2013; De Wekker and Kossmann, 2015; Serafin et al., 2018). Over the Alps, for instance, these mechanisms may commonly manifest as Alpine pumping (Lugauer and Winkler, 2005; Weissmann et al., 2005; Graf et al., 2016). Furthermore, during daytime scalar quantities such as heat, moisture, and aerosols typically generated in populated valleys may be exported outside of the mountain range owing to a combination of both buoyantly and

mechanically driven pathways. Such pathways include mountain venting processes, in which case the combined effects of valley and slope flows determine the fate of the scalars in question (De Wekker and Kossmann, 2015). Mountain-plain wind circulations, acting across longer, cross-chain scales, represent an extreme case of such a pathway (Goger et al., 2019). When regional winds across a mountain range are significant, further downwind export of these scalars may be recorded (Kossmann et al., 1998; Adler and Kalthoff, 2014; Lang et al., 2015; Pal and Lee, 2019), ultimately affecting the downwind ABL structure.

It is therefore essential to provide a meaningful diagnostic for expressing the extents of local, radiative surface-connected forcing on one hand, and that of an entire mountain range on the other hand. While the former dictates the evolution of a CBL, the latter is ultimately responsible for the vertical extent of the MoBL, thus permitting the expression that the MoBL is a parent layer to the CBL over mountains (De Wekker and Kossmann, 2015). Mindfulness of this discrepancy is critical (Lehner and Rotach, 2018), as any misrepresentation of the depths of the two layers may lead to substantial errors concerning

pollutant transport and heat exchange (Gohm et al., 2009; Ketterer et al., 2014; Leukauf et al., 2017; Giovannini et al., 2020) and associated drag effects on regional and global wind systems in numerical weather prediction models (Jiang et al., 2008; Vosper et al., 2018).

For the purpose of exploring CBL evolution in time and space, the governing prognostic equation describing the CBL depth, taken here as the height of a potential temperature inversion at a height $z_i$ above ground, may be expressed as (Stull, 1988;

Kossmann et al., 1998; Kalthoff et al., 2020):

$$\frac{\partial z_i}{\partial t} + u_{z_i}\frac{\partial z_i}{\partial x} + v_{z_i}\frac{\partial z_i}{\partial y} = \frac{1}{\Gamma}\left[\frac{\partial \Delta_\theta}{\partial t} + \frac{\overline{w'\theta'}_0 - \overline{w'\theta'}_{z_i}}{z_i}\right] + w_L, \tag{1}$$

where $u_{z_i}$ and $v_{z_i}$ are the horizontal wind speed components at the inversion, $\Gamma$ is the static stability above the inversion, $\Delta_\theta$ is the potential temperature jump at the inversion, $\overline{w'\theta'}_0$ and $\overline{w'\theta'}_{z_i}$ are the kinematic sensible heat fluxes at the surface and inversion, respectively, and $w_L$ is the large-scale subsidence above the inversion, with subsidence $w_L < 0$ by definition. Using

conventional observational systems such as radiosoundings or remote sensing systems (Banta et al., 2013; Emeis et al., 2018; Kottmeier et al., 2021), only a handful of terms in Eq. 1 are usually obtainable along single columns, resulting in a simplified,



one-dimensional CBL growth framework, known also as the encroachment model (Stull, 1988; Batchvarova and Gryning, 1991):

$$\frac{\partial z_i}{\partial t} = \frac{\overline{w'\theta'}_0}{\Gamma z_i} + w_L. \tag{2}$$

Despite its disregard of the entrainment zone dynamics, it nonetheless explains well the bulk evolution of CBL in time and over HHF terrain (Stull, 1988). On the other hand, neglecting the horizontal advection terms over complex terrain, as they are extremely challenging to appropriately sample, may be unjustified (Kossmann et al., 1998).

The degree to which the MoBL or the CBL top can be appropriately identified has been largely dependent on the measurement platform utilized for determining their respective depths (Emeis et al., 2008, 2018). The two main types of instruments

capable of estimating their respective depths may be divided into in-situ and remote sensing platforms, with radiosondes as the most reliable type of the former platform (Seidel et al., 2010, 2012), and Doppler lidars (Dang et al., 2019; Ritter and Münkel, 2021) as well as microwave radiometers (Cimini et al., 2013) as the most popular types of the latter platform. Ceilometers and Doppler as well as aerosol lidars have been reliably used to estimate CBL and MoBL depths in the past. Their reliance on abundantly aerosol-laden air for the estimation of CBL and MoBL depths is known to introduce ambiguity in interpre-

tation (De Wekker et al., 2004). This holds particularly for determination methods relying on vertical profiles of backscatter coefficients from ceilometers and aerosol lidars (Emeis et al., 2008). Furthermore, ceilometers are unable to reliably detect CBL depths whenever vertical layers are nearly homogeneously distributed, thereby requiring either adjusting the detection parameters or manually discarding such cases entirely (van der Kamp and McKendry, 2010). Although more complex detection methods have been developed to account for such scenarios (Eresmaa et al., 2012), situations specific to complex terrain, namely non-locally advected layers associated with higher backscatter coefficients than those encountered in the CBL below,

remain more difficult to tackle.

Up to now, estimates of CBL and MoBL depths obtained from vertically pointing systems have suffered from a detrimental lack of spatial representativeness, especially over mountainous terrain due to the non-local pathways described previously. Doppler lidars aligned in a coplanar fashion, a methodology on the rise in recent years, has been able to partly overcome

the issue of representativeness, while also providing novel perspectives into dynamics governing CBL and MoBL evolution, particularly within valleys (Hill et al., 2010; Wildmann et al., 2018; Adler et al., 2020, 2021d; Haid et al., 2020, 2022; Babić et al., 2021, hereafter B21). Nonetheless, a method relying on coplanar-deployed lidars capable of retrieving spatially rich information of CBL and MoBL depths in a routine manner, with an emphasis on turbulence characteristics rather than on aerosol abundance, is still lacking.

An example of a such a methodology related directly to turbulence intensity is turbulent Doppler spectrum broadening (hereafter spectrum width), which has found multiple uses for the past four decades (Istok and Doviak, 1986; Heo et al., 2003; Wildmann et al., 2019). The possibility of disentangling the contribution of shear and buoyancy when analyzing Doppler spectra is a significant advantage of spectrum width over radial velocities, for which the turbulence origin information is not crucial (Smalikho et al., 2005; Wildmann et al., 2019). Furthermore, the inclusion of the Doppler spectrum width to lidar-based





velocity variances may provide a significantly more reliable picture of both unstable and particularly stable boundary layers impacted by wind turbine wake over complex terrain (Wildmann et al., 2020).

This study attempts to remedy the lack of such a methodology, by fulfilling the following three goals: i) to describe a novel method for determining the CBL depth in a valley; ii) to analyze a highly spatio-temporally resolved distribution of CBL depths $z_i$ across a roughly 20 km wide Alpine valley; iii) to identify prevalent MoBL regimes at specific times of day. To accomplish this, we utilize spectrum width sampled by three Doppler lidars across the Inn Valley, Austria, thereby fulfilling the criterion of relying primarily on turbulence properties. As just one of many options for determining $z_i$, spectrum width is ideally suited for our goal of determining CBL depth, due to its relation to turbulence intensity and availability in regions close to the slopes where traditional parameters, such as true vertical velocity, may be difficult to obtain. In this study, we explore four Intensive Observation Periods (IOP) that took place during a single week in August 2019 during the "Cross-valley flow in the Inn Valley investigated by dual-Doppler lidar measurements" (CROSSINN) campaign (Adler et al., 2021d).

Recent findings based on the CROSSINN measurements revealed that the daytime MoBL in this slightly curved portion of the Inn Valley is characterized by a cross-valley vortex (CVV) during synoptically undisturbed conditions when a strong upvalley flow develops, a phenomenon resulting from a height-dependent force imbalance between the centrifugal force on one hand and the pressure gradient force on the other hand (B21). Such a force imbalance with height was shown to generate a pronounced clockwise vortex when looking upvalley, whose vertical extent filled the entire cross-valley transect.

The present study is organized as follows: Section 2 introduces the sites and the analyzed data sets, as well as the processing of spectrum width and the detection method of CBL and MoBL depths; Section 3 provides the results of the analysis, focusing first on the temporal evolution of terms in Eq. 2 above the Inn Valley floor, then leading up to the introduction of distinct MoBL evolutionary stages and finally a novel exploration of cross-valley spatial variability of terms in Eq. 2; Section 4 lastly provides an overview of the study with potential implications and future outlook.

## 2 Investigation area, instrumentation, methodology

### 2.1 Site details

CROSSINN took place in the lower Inn Valley, approximately 18 km to the northeast of Innsbruck. The valley in this region is approximately 10 to 15 km wide crest-to-crest, roughly 1630 m deep (Adler et al., 2021d), while the valley floor is flat, in the cross-valley direction, for roughly 2 km (Fig. 1). The valley floor is dominated by a mixture of residential and cultivated areas, while the neighbouring valley sidewalls are mostly forested. The southern valley sidewall is characterized by a number of tributary valleys exiting nearly perpendicularly into the Inn Valley. Although the northern sidewall possesses no similar tributaries, it is characterized by a relatively flat, nearly 2-km wide plateau, extending approximately 300 m above the valley floor (AVF). The valley floor itself is located at an altitude of 546 m above mean sea level (MSL) in the investigation area.



## 2.2 Instrumentation details

We refer the reader to Adler et al. (2021d), their Table 3, as well as to B21, for a detailed overview of technical characteristics, scan types, and scan configurations concerning the remote sensing instrumentation. Here we focus on describing only a subset of the overall CROSSINN instrumentation of relevance for the present investigation. Note that all reported times are in UTC (local time LT = UTC + 2).

### 2.2.1 Windcube Doppler lidars

Three Windcube WLS200s lidars, otherwise components of the KITcube measurement facility (Kalthoff et al., 2013), were deployed at three sites along a transect oriented in the cross-valley direction (Fig. 1). The reader can find the overview of the investigation area in Adler et al. (2021d) and B21. The accurate positioning of the three Doppler lidars, in an area of heterogeneous complex terrain, yielded three scanning planes which were offset by at most 120 meters in the along-valley direction. The three lidars performed back-and-forth range height indicator (RHI) scans at an azimuth angle of 158.15°. The scanning region was chosen as a compromise between achieving a sufficiently large overlap region while still ensuring a fairly high temporal resolution between scans. As such, each RHI scan lasted 1 min, providing radial velocity $v_r$, carrier-to-noise ratio (CNR), and Doppler spectrum broadening at a physical resolution of 100 m, with overlapping range gate distance of 50 m, and a temporal resolution of 0.25 s. For more insight into the post-processing of these RHI scans, we refer the reader to B21.

### 2.2.2 Flux towers

The siting of CROSSINN instrumentation was primarily influenced by the locations of the longterm i-Box flux towers (Rotach et al., 2017). Specifically, the three flux towers of interest for the present study were CS-VF0, CS-SF1, CS-NF27, where CS stands for "core site", VF for "valley floor", NF and SF for "north-facing" and "south-facing", respectively, and the digit represents the local slope angle in degrees. These three sites will in order hereafter be referred to as Kolsass, Mairbach and Hochhäuser. More details concerning these sites and the data post-processing procedure can be found in Stiperski and Rotach (2016), B21, and Lehner et al. (2021). For this study we have chosen to select horizontal wind speed and direction (8.7-m AGL) only at the CS-VF0 (Kolsass) site, while the surface sensible heat flux from all three sites will be used to distinguish unstable from stable ABL states. We refer the reader to Rotach et al. (2017) for more details concerning the setup of each of these three sites.

### 2.2.3 Radiosoundings

GRAW DFM-09 radiosondes were launched at discrete times from Kolsass to provide highly resolved in-situ measurements of base thermodynamic variables during the four IOPs examined in this study. The temporal resolution of each sounding was 2 s, resulting in an average height interval of approximately 5-10 m. For a complete overview of each IOP and exact launch times, we refer the reader to Adler et al. (2021d), their Fig. 3.





### 2.2.4 Aircraft measurements

The D-FDLR Cessna Grand Caravan 208B (German Aerospace Center, DLR) performed four along-valley flights during both IOPs 2b and 4 (Adler et al., 2021d; B21). The aircraft sampled in-situ measurements of all three wind speed components, temperature, as well as specific humidity at a sampling rate of 100 Hz, with the METPOD sensor assembly (Mallaun et al., 2015). We analyze the flight legs flown at various levels within developing CBLs to validate features in the spectrum width sampled by the three Windcube lidars that are reminiscent of convection. Figure 1 illustrates the orientation of each analyzed along-valley flight leg segment.

### 2.3 Methodology

#### 2.3.1 Coplanar retrieval method

The focus of the CROSSINN campaign was the study of cross-valley flow, via the application of the coplanar retrieval method, known also as the dual-Doppler method (Calhoun et al., 2006; Hill et al., 2010; Newsom et al., 2015; Whiteman et al., 2018a, b; Wildmann et al., 2018; Peña and Mann, 2019; Adler et al., 2020, 2021d; B21; Haid et al., 2020, 2022; Wittkamp et al., 2021). During CROSSINN, this type of a retrieval method was applied to overlapping RHI scans performed by the three Doppler lidars. The alignment of the three RHI scans from each Doppler lidar, along with their synchronized start and end times, made it possible to retrieve $\mathbf{v}_v$, the two-dimensional projection of the three-dimensional wind field onto a vertical cross-section. In this study, retrievals of $\mathbf{v}_v$ are performed onto a 8 km wide and 5 km tall Cartesian mesh with a lattice width $\Delta l$ equal to 50 m (Adler et al., 2021d; B21). The retrieved $\mathbf{v}_v$ is rejected whenever the intersection angle of the original lidar beams was outside of the interval $[30°, 150°]$, simultaneously for all three lidar pairs, resulting in regions with no cross-valley wind information near the Inn Valley sidewalls. Such a wide elliptical region has been discarded due to potentially large uncertainties of $\mathbf{v}_v$ owing to a near co-linearity of each lidar beam pair. In the present study we will only use the vertical wind speed component of the retrieved $\mathbf{v}_v$ wind field, denoted hereafter as $w$.

#### 2.3.2 Spectrum width post-processing and merging

Turbulent Doppler spectrum broadening $\sigma_t^2$ can be defined as follows (Smalikho et al., 2005; Wildmann et al., 2019):

$$\sigma_t^2 = \sigma_{sw}^2 - \sigma_0^2 - \sigma_s^2 - E, \tag{3}$$

where $\sigma_{sw}^2$ is the measured spectrum width, $\sigma_0^2$ equals the spectrum width at constant wind speed within the sampling volume, $\sigma_s^2$ is the spectrum width due to shear effects, and $E$ is the random error. During CROSSINN, no raw spectra data were stored primarily due to limitations in data storage capacity. Therefore, we resort to the spectrum width values, as outputted by the WLS200s internal software package Windforge. Although these products have $\sigma_0^2$ accounted for, they haven't been corrected for shear effects and random errors (Leosphere, personal communication). As we will demonstrate in the following sections, this incompleteness does not affect the bulk contrasts between turbulent air within the CBL and less turbulent air aloft. Since





the post-processing of these outputted spectrum widths requires a different set of criteria for removal of bad data than that for radial velocities (B21), in the following we describe these steps in more detail.

To facilitate analyses, for each 1-min RHI scan we first computed the average spectrum width and CNR at each grid point of the coplanar-retrieval mesh which fell within the radius of influence used already for the coplanar retrieval. We determined that for the broad range of conditions encountered during CROSSINN, a more strict CNR criterion was required in case of spectrum width, compared to radial velocities, to filter out general outliers within an RHI scan. Specifically, we filtered out all values with CNR less than -24 dB, compared to -28 dB used for radial velocities. We encountered two additional issues which warranted closer inspection and handling. First, the presence of ubiquitous oscillatory artefacts in the Mairbach scan (Appendix A), which persisted throughout the campaign, were treated satisfactorily owing to the regularity of the oscillations. Second, the range of spectrum width values at Hochhäuser and Mairbach were respectively lower and higher compared to Kolsass. This disparate range of spectrum width values among the three systems were accounted for (Appendix B), by applying daily offsets to the Hochhäuser (Fig. 2a) and Mairbach (Fig. 2c) RHI scans in order to bring their median values as close as possible to that of the Kolsass RHI scans (Fig. 2b). Due to these two additional post-processing steps, we will hereafter denote the magnitude of spectrum width in arbitrary units (a.u.).

Although the three lidars scanned each section of the valley cross-section at slightly different times within each 1-min period, the agreement in both location and spatial extent of the most prominent spectrum width features among the three RHIs is encouraging. This holds particularly for the features found 1500 m south and 1100 m north of Kolsass. This encouraging degree of stationarity within a single 1-min RHI scan suggests, in turn, a potential benefit from merging together the three separate spectrum width fields. Merging scans in this manner can provide detailed and frequent snapshots of the turbulence state across the whole valley. This merging was done via simple averaging of the three post-processed RHI scans at each grid point of the coplanar retrieval mesh, for each 1-min combination. In the remainder of the study we focus only on the region between the Hochhäuser and Mairbach Doppler lidars (Fig. 3).

## 2.4 Validation of spectrum width against vertical velocity

Depicted in Fig. 3a is an early afternoon period on IOP 4, one of the most convective days sampled during CROSSINN. The developing CBL was characterized by intense surface-driven convection and weak along-valley flow (B21, their Fig. 8c). Note that the CVV has not yet begun to form at this time of day (B21). At this moment, the CBL was characterized by two narrow thermal updrafts, located approximately 1700 m south and 1100 m north of Kolsass, respectively, both reaching roughly 1000 m AVF. Unfortunately, there is no vertical velocity information close to the surface, due to the intersection angle region constraint explained earlier (Section 2.3.1). Lack of vertical wind information there discourages any worthwhile investigation of CBL evolution, especially during morning hours when convective features are shallow.

This disadvantage of the coplanar retrieval methodology can be overcome by considering the merged spectrum width field in addition to the vertical wind (Fig. 3b), since this is derived directly from the RHI scans. At first glance, the similarity between the shape and location of elevated spectrum width regions and the updrafts (Fig. 3a) is striking. More importantly, Fig. 3b demonstrates that the two updrafts were both attached to the surface, a characteristic one cannot infer from Fig. 3a.



Nonetheless, one must keep in mind that the lidar cross-valley transect is also sampling such convective features as they get advected through the investigation area. For instance, a convective structure being advected by the along-valley flow could therefore, despite its expected lifetime of 5-10 minutes in a well developed convective environment, pass through our transect over a much shorter period of time. As a result, instances of low turbulent regions capped by more turbulent air, such as the one over the plateau about 2000 m north of Kolsass, suggest a transient, non-locally generated thermal detached from the

plateau. Similarly, the thermal in question may still be attached to the plateau, but simply vertically tilted. As a result, inferring instantaneous CBL depths for such cases, as opposed to a temporal averaging approach where for instance hourly averages are computed, may be misleading. Keeping in mind that the showcased spectrum width field was previously manipulated during post-processing (Section 2.3.2), it is nonetheless clear that the turbulence levels within the two updrafts (greater than 0.7 a.u.) were nearly double in magnitude compared to the upper valley atmosphere (predominantly 0.45 a.u.). We observed

background values outside of the boundary layer rarely to drop below 0.38 a.u. (not shown). This contrast between surface-driven convection adjacent to the surface, and low-turbulence regions aloft, as inferred from spectrum width variability in space, lends itself as a tentatively useful tool for tracking the CBL evolution in both time and space.

### 2.4.1 Bottom-up exceedance threshold method for CBL depth $z_i$ determination

As previously highlighted, the contrast between relatively higher spectrum width values within the CBL compared to those

230 aloft may be utilized to determine the depth of the CBL. We emphasize that this determination was performed on each vertical column of spectrum width in the cross-valley transect, computed as an hourly mean from sixty 1-min instances. Hereafter we apply the method to 1-hr windows shifted 10 minutes forward in time in order to improve temporal resolution and increase visual detail.

We found that within the CBL the 1-hr mean spectrum width typically gradually decreases with height until it reaches upper

level ambient values of around 0.4 a.u. (Section 3.3), rather than exhibiting abrupt changes near the CBL top that could hint at the entrainment zone. Such gradually decreasing tendency is instead reminiscent of a vertical profile of vertical velocity variance, a common parameter used to infer CBL depths from vertically pointing lidars (Adler and Kalthoff, 2014, their Fig. 2). Therefore, we resorted to a bottom-up exceedance threshold method for determining the CBL depth $z_i$, leaving us with the task of defining the most appropriate threshold value. For simplicity we will report the CBL depth as $z_i$, though it may

not always correspond to the more traditional definition of CBL depth $z_i$ representing the height of the temperature inversion capping the CBL.

To accomplish' $z_i$ estimation, we calibrated the bottom-up method against the bulk-Richardson number method for determining ABL depths from virtual potential temperature $\theta_v$ profiles sampled by radiosonde (Kalthoff et al., 2020; Ladstätter, 2020). The bulk-Richardson method relies on targeting the height at which the bulk-Richardson number falls below a widely adopted

threshold value of 0.25 (Vogelezang and Holtslag, 1996; Herrera-Mejía and Hoyos, 2019). The bulk-Richardson number $R_B$ equals:

$$R_B = \frac{gz}{\theta_{v,0}} \frac{\theta_v - \theta_{v,0}}{u^2 + v^2}, \qquad (4)$$



where $g$ is the gravitational acceleration, $z$ the height AVF, $\theta_v$ the virtual potential temperature profile, $\theta_{v,0}$ the surface virtual potential temperature profile, and $u$ and $v$ the two horizontal wind speed component profiles. We applied the $R_B$ method only to the radiosonde launches performed between 9 and 13 UTC, as this period corresponds to the convectively driven phase (B21; Lehner et al., 2021) when the bottom-up method was observed to perform the best. A possible weakness of the $R_B$ method is its reliance on the accuracy of $\theta_{v,0}$ in highly convective summer conditions. Specifically, owing to the superadiabatic character of the surface layer (Zhang et al., 2014), an unusually deep CBL may be identified, which may have a negative effect on our calibration of the bottom-up method. To establish this tentative influence, we have applied the $R_B$ method to the 9, 11, and 13 UTC radiosonde launches for all ten CROSSINN IOPs, by varying the number of near-surface data points involved in computing $\theta_{v,0}$.

To identify the optimal spectrum width threshold for the bottom-up method, we computed $z_i$ from all 1-hr spectrum width profiles above Kolsass corresponding to the radiosonde launches for a range of thresholds between 0.38 and 0.47 a.u. These $z_i$ were contrasted against those obtained with the $R_B$ method, for a range of $\theta_{v,0}$ values obtained as averages from the lowest three up to lowest twelve data points of the vertical profile. Given the average spacing between two sounding data points of ten meters, this corresponds to a layer on average between 30 and 120 m thick. For each matching pair of the $z_i$ values from the two methods, the root mean squared error $RMSE$ was computed (not shown) as the most representative statistical parameter to convey the scatter between the two $z_i$ estimates. Figure 4, resulting from the choice of 0.44 a.u. threshold and nine data points for the computation of $\theta_{v,0}$, shows the pairing exhibiting the lowest $RMSE$ of all combinations, leading us to choose 0.44 a.u. as the final spectrum width threshold. Justified by Fig. 4 and to avoid unreasonably deep $z_i$, we have imposed an upper limit of 1100 m depth above local topography, beyond which all $z_i$ values have been labeled as erroneous and discarded in the analyses that follow. This specific height threshold was chosen with respect to the negligible chance of a CBL reaching a depth greater than 1100 m (Fig. 4), not just over Kolsass, but in the rest of the cross-section as well. No radiosonde-derived $z_i$ were removed with this filter. As a result, Fig. 4 highlights the rather shallow nature of summer CBLs in the Inn Valley not exceeding more than a kilometre in depth. Seasonality, expressed via progressively lower sensible heat fluxes, is also evident, wherein deeper CBLs occurred during earlier IOPs in August while shallower CBLs were more common during later IOPs in September.

## 2.5 Validation of spectrum width features in space: An airborne eddy-covariance perspective

The calibration of the bottom-up method with the $R_B$ method, despite being a success, left the representativity of the cross-valley transect with respect to the rest of the investigation area unaddressed. Additionally, it remains unclear whether instantaneous, 1-min spectrum width features, such as the turbulent thermals visible in Fig. 3b, can indeed be equated with regions of convectively-driven turbulence. Given the lack of systematic and routine in-situ measurements of turbulence parameters other than at the surface, this validation cannot be performed in a manner similar to the radiosondes earlier. Fortunately, the DLR Cessna aircraft flew a number of flight legs both inside and outside of the CBL on IOPs 2b and 4 (Fig. 1), while sampling at a sufficiently rapid rate to allow a robust computation of spatio-temporal averages (Tab. 1). We will explore three different flight legs flown at various stages of CBL development: i) a late morning flight leg flown near the top of a shallow CBL on IOP 2b



(Fig. 5a); ii) a late morning flight leg flown within a more turbulent and relatively deeper CBL on IOP 4 (Fig. 5b); iii) a late afternoon flight leg flown slightly above an apparently mature CBL on IOP 2b (Fig. 5c), given the time of day when the leg was flown. To assess the turbulent state and fidelity of corresponding spectrum width features through which the aircraft flew, we

will consider the in-situ perturbations of vertical velocity $w'$ (Fig. 6a-c), potential temperature $\theta'$ (Fig. 6d-f), specific humidity $q'$ (Fig. 6g-i), as well as instantaneous kinematic heat flux $w'\theta'$ (Fig. 6j-l) and instantaneous kinematic moisture flux $w'q'$ (Fig. 6m-o). Perturbations are computed as linearly detrended values only for the straight leg segments. The standard deviation of the aircraft height along the straight leg segment amounted to 2, 9, and 6 m, while the average aircraft velocity equaled 68, 63, and 66 m s$^{-1}$, for each of the three flights, respectively. Although the horizontal leg segments were slightly curved to account

for the valley curvature in this part of the Inn Valley (Fig. 1), we nonetheless assume a negligible impact of aircraft turning on the quantities scrutinized here, considering we do not focus on horizontal velocity components.

We begin with the late morning flight performed on IOP 2b, with the relevant straight leg segment flown between 08:24 and 08:27 UTC at an average altitude of approximately 270 m AVF (Figs. 5a, 6). Unlike the example introduced by Fig. 3, the valley atmosphere lacked any vigorous and deep thermal activity at this time, with most of the surface-driven convection

situated above the plateau. Furthermore, three layers of elevated spectrum width are found at heights of approximately 700, 900, and 1250 m AVF, most likely due to wind shear present in the downvalley flow at this time (B21, their Fig. 8a). Since the aircraft flew at a height where, based on spectrum width magnitude, turbulence intensity was fairly weak, we can expect to observe weak perturbations and covariances sampled by the aircraft. Indeed, the majority of the most vigorous $w'$ and $q'$ were concentrated between the 6th and 10th kilometer of the segment (Fig. 6a,g), resulting in peak positive $w'q'$ there as well (Fig.

6m). On the other hand, the most energetic $\theta'$ perturbations were associated with mesoscale gradients, evident by the distinct jump between the 2nd and 6th km of the segment (Fig. 6d). Nonetheless, the positive signs of both $w'\theta'$ and $w'q'$ agree with the textbook expectancy of these covariances to be positive inside the CBL (Stull, 1988; Moeng and Sullivan, 1994; Schmidli, 2013; Wulfmeyer et al., 2016), suggesting that the aircraft has indeed flown through the CBL, but may not have remained within it for the entire duration of the segment.

Having been flown at an average elevation of 263 m AVF, the aircraft encountered markedly stronger turbulence levels during the late morning flight on IOP 4, between 09:35 and 09:37 UTC (Figs. 5b, 6). As a result, the vertical velocity perturbations $w'$ shown in Fig. 6b were larger than on IOP 2b (Fig. 6a), with more equidistant perturbations of the same order of magnitude between -2 and 3 m s$^{-1}$. Although the range of $\theta'$ and $q'$ were comparable to those on IOP 2b, they corresponded to a greater degree to the $w'$ features, therefore resulting in larger magnitudes of the respective covariances as well (Fig. 6k,n; Tab. 1). In

particular, the fairly regular positive $w'q'$ excursions between kilometers 0 and 6 suggest nearly equidistant ejections of moist air from the surface layer to the level of the flight leg. Overall, the more turbulent state of the CBL on IOP 4 is also reflected in the vertical velocity variance $\langle w'^2 \rangle$ values being more than double in magnitude of those on IOP 2b (Tab. 1). Despite this leg having been flown a little over an hour later than on IOP 2b, the positive signs of both $\langle w'\theta' \rangle$ and $\langle w'q' \rangle$ once again confirm the leg was flown within the CBL.

To ascertain whether the spectrum width approach of studying the CBL is viable not just for developing CBLs, but also in case of mature ones potentially not coupled to the surface anymore, we also explore the third flight leg, having taken place on



late afternoon of IOP 2b, between 14:19 and 14:21 UTC at an average altitude equal to 1084 m AVF (Figs. 5c, 6). Characterized by low turbulence levels over the valley floor, most of the supposedly non-locally convectively-driven turbulence at this time occurred above the plateau, between 500 and 1100 m AVF (Fig. 5c). However, there seems to have been a disconnect between the turbulence in the surface layer above the plateau between 1000 and 2000 m north of Kolsass, and those higher turbulence regions aloft. Taking into account that the upvalley flow at this time was stronger than the downvalley flow six hours earlier (Figs. 5a), we conclude that in case of sufficiently tall thermals in a mature CBL, their advection by the mean wind through the lidar transect should be an important factor to consider. Furthermore, given the fact that the flight leg was located near the upper boundary of most turbulent thermals, we should not expect to encounter similar relationship in signs of the covariances compared to the two flight legs examined earlier (Figs. 5a,b). Since the leg was flown in the vicinity of a mature CBL, the perturbations $w'$, $\theta'$, and $q'$ (Fig. 6c,f,i) reflect evidence of thermals, specifically four discrete ejections ($w' > 0$) of potentially colder ($\theta' < 0$) and moister ($q' > 0$) air from the CBL to the level of the aircraft. In Fig. 6c,f,i these four signatures may be found approximately at kilometers 2, 4, 5 and 8, respectively. This is particularly evident from the deviations of the specific humidity from ambient, upper-valley levels of humidity. As a result, $w'\theta'$ at the locations of the four discrete thermals is primarily of negative sign (Fig. 6l), while $w'q'$ is primarily of positive sign (Fig. 6o). Spatio-temporal averages of these two covariances (Tab. 1) reflect this relationship between the two, suggesting that the aircraft most likely flew within the CBL entrainment zone or slightly above it, due to the negative sign of $\langle w'\theta'\rangle$ (Stull, 1988; Wulfmeyer et al., 2016). Lastly, the presence of such discrete thermals along the entire leg, spaced fairly evenly apart, suggests that the entire plateau supported a similar CBL structure, fortifying the representativeness of our single cross-valley lidar transect.

## 3 Spatio-temporal determination of MoBL behaviour in the investigation area

Having calibrated the bottom-up method to accommodate the range of conditions specific to the Inn Valley during summertime, as well as adequately validated the spectrum width features with in-situ measurements, in the remainder of the study we focus our attention on four IOP events (Adler et al., 2021d): IOPs 2a (8 August), 2b (9 August), 3 (11 August), and 4 (14 August). These successional IOPs have several favourable characteristics which help us minimize the degrees of freedom of the research questions listed in Section 1. First, they span a single week, from 8 to 15 August, therefore allowing us to disregard seasonality, and by extension, possible changes in varying sensible and latent heat fluxes between the IOPs. Second, none of the four IOPs experienced prolonged periods of cloud cover during daytime which would have, due to their intermittency, made heat flux comparisons more difficult and inhibited lidar measurements. However, with the exception of IOP 2b, all other IOPs experienced a sustained cloud cover during night (not shown), disrupting the radiation balance before sunrise and thus helping explain some of the differences between observed sensible heat fluxes among the four IOPs. Third, they took place within a time window short enough to allow us to assume a stationary state of the land use in the investigation area, thus constraining the regional relationships between sensible and latent heat fluxes driving the MoBL. Fourth, these IOPs experienced a substantial range of different large-scale forcings, permitting us to gain deeper insight into different mechanisms affecting CBL structure.



### 3.1 Temporal variability of parameters impacting CBL development

We first explore the evolution of traditional parameters involved in studying CBL growth development. We focus on examining near-surface conditions at the valley floor only, with the aim of characterizing the along-valley wind, the observed $z_i$ variability, and of relating this variability to the temporal evolution of terms in Eq. (2). Due to the surrounding ridges, on the day of IOP 3 (11 August) Kolsass experienced local sunrise and sunset at approximately 04:54 UTC and 17:40 UTC, respectively. In the analyses to follow, we will assume that these times were valid for the entire considered week.

As established recently in B21, the pattern of near-surface wind evolution at the floor of the Inn Valley exhibited typical diurnal trends, namely the prevalence of downvalley flows at night, and relatively stronger upvalley flows at day (Fig. 7a,b). However, of the four targeted IOPs, only IOP 2b exhibited a pronounced morning downvalley flow. This deviation of IOP 2b during nighttime could be due to synoptic foehn influence, as the foehn probability diagnostic (Plavcan et al., 2014) indicated a high probability of foehn on this day (Adler et al., 2021d, their Fig. 5c). During daytime, all IOPs experienced an upvalley

flow, but to varying extents. While the strongest and longest upvalley flow took place on IOP 2a, a weaker upvalley flow with a later than usual onset occurred on the following day, during IOP 2b. Daytime upvalley flows typically began around 10:00 UTC, ceasing fully by 21:00 UTC. These time frames correspond to the previously determined CVV durations (B21). Fully developed CVVs spanning the entire valley depth were present only on IOPs 2a and 4. Although IOP 3 based on Fig. 7 seems to have behaved in a manner similar to other IOPs, following analyses will prove how consideration of only near-surface

parameters may often be misleading when it comes to complex terrain flows. As expected, the sensible heat flux for all IOPs (Fig. 7c) similarly became positive at approximately 05:30 UTC, while changing sign again fairly early compared to local sunset, around 13:30 UTC. Such an early sign change is a result of relatively larger latent heat fluxes in the Inn Valley, owing to cultivated land use and patchy forested regions (Lehner et al., 2021). While IOPs 2a, 3 and 4 exhibited the highest sensible heat fluxes at 09:00 UTC of around 100 W m$^{-2}$, IOP 2b stood out, as its peak value occurred somewhat later at 12:30 UTC,

for which we hypothesize that the foehn influence may have sufficiently altered the otherwise expected behaviour of energy partitioning near the surface of the Inn Valley.

Next we turn our attention to the CBL development, initially above Kolsass only. Starting at approximately 06 UTC, the CBLs gradually deepened, but with varying growth rates (Fig. 7d). By 13 UTC, IOP 2b has evidently established itself as a shallow CBL case, not exceeding 300 m in depth. On the other hand, CBLs during the other three IOPs reached approximately

800 to 1200 m AVF. Such a CBL growth behavior corresponded fairly well with the prevalent positive sign of the sensible heat flux $H$ at Kolsass between 06 and 13 UTC (Fig. 7c), assuming $H$ is the sole driver of $z_i$. After 13 UTC, the $z_i$ estimates among the IOPs diverged considerably. Some inconsistencies have arisen, for instance the shallowest $z_i$ despite the largest $H$ on IOP 2b between 11 and 13 UTC. To offer an explanation for this, we consider next the two opposing factors of simplified one-dimensional CBL growth (Eq. 2), namely stability $\Gamma$ and subsidence $w_L$.

Upper-level stability $\Gamma = \Delta\theta/\Delta z$ was computed for the radiosonde launches between 06:00 and 13:00 UTC, as the lapse rate in a 700 m deep layer above the local $z_i$ above Kolsass at the time of launch. Experimenting with various layer thicknesses offered no evidence of change of the dominant $\Gamma$ tendencies with time (not shown). The 06:00 UTC launches, conducted




shortly after sunrise, were included given the fact that CBL growth models take $\Gamma$ as an initial condition, while also assuming
it remains constant over the course of the day. As clearly depicted by Fig. 7e, this assumption is violated in the Inn Valley
because the layer above the CBL is still a part of the MoBL. Rather, the free tropospheric stability, above approximately 3000
m AVF, was observed to remain steady during daytime on all IOPs (not shown), suggesting that the influence of the MoBL at
the Alpine scale may cease beyond 3000 m AVF. One possible explanation for the overall shallowest CBL encountered on IOP
2b could be the largest 06 UTC stability $\Gamma$ of nearly 7 K km$^{-1}$. Even more so since, unlike the other three IOPs, particularly
IOPs 2a and 4, $\Gamma$ on IOP 2b remained fairly high until 11:00 UTC. On the other hand, the deeper CBLs on IOPs 2a and 4 (Fig.
7d) experienced less resistance during their deepening, by growing into progressively less stable air aloft, as indicated by their
$\Gamma$ values decreasing from roughly 4 to 1 K km$^{-1}$. During IOP 3, $\Gamma$ stayed fairly constant in the range 4.5 - 6.5 K km$^{-1}$ (Fig.
7e), however it did not exhibit a typical decrease over the course of the day as was the case with the other three IOPs.

Lastly, we aim to establish the main temporal characteristics of subsidence $w_L$, here calculated as the average coplanar-
retrieved vertical velocity in a layer 100 m thick directly above local $z_i$ above Kolsass (Fig. 7f). Although subsidence values
exhibit large variability among the IOPs, they do not exhibit any consistent trend during daytime, with strongest subsidence
reaching -0.25 m s$^{-1}$ on average. Despite being based on 1-hr averages, these $w_L$ are still larger than typical subsidence values
encountered over flat, horizontally homogeneous terrain (Avissar and Schmidt, 1998; Blay-Carreras et al., 2014; Pietersen
et al., 2015), even over a single mountain range (Kalthoff et al., 1998; Kossmann et al., 1998). Subsidence values of the order
of magnitude found in the observations shown here correspond to the ones encountered in highly resolved daytime valley flow
simulations conducted by Serafin and Zardi (2011). Interestingly, between approximately 07:00 and 10:00 UTC, both IOPs
2a and 2b experienced comparably intense subsidence, between -0.2 and -0.3 m s$^{-1}$. During this time frame, both $H$ and
$z_i$ between the two IOPs behaved similarly as well. After 10 UTC the CBLs on these two IOPs began to diverge in depth,
suggesting that after 10:00 UTC, the far more stable air above the CBL on IOP 2b was the main factor opposing the CBL
growing any deeper than 300 m. It is important to emphasize that the deeper well-mixed boundary layer found later on IOP
2b, specifically at 16:00 UTC with a depth of 900 m, took place during time when the sensible heat flux at the surface already
became negative (Fig. 7c). In the remainder of the study, we will refer to these afternoon boundary layers, due to the broad
variety of the origin of their turbulence, simply as a well-mixed boundary layer, rather than the MoBL which we are not
sampling entirely. As shown by theoretical derivations performed by Ouwersloot and Vilà-Guerau de Arellano (2013b), the
effects of stability and subsidence do not add linearly, demonstrating that the effect of subsidence becomes more pronounced
with weaker stability (Ouwersloot and Vilà-Guerau de Arellano, 2013a). This seems to have been the case for IOP 2a only
until 10:00 UTC or so, after which the CBL suddenly gained depth despite a stronger subsidence around 12:00 - 13:00 UTC
than previously (Fig. 7f). Once again we emphasize the high probability of foehn on IOP 2b, hinting at a possible inability
of our bottom-up method to differentiate between convectively-driven turbulence and foehn-induced turbulence as was the
case in this IOP. Based on Fig. 7 we hypothesize that upper-valley stability $\Gamma$ may serve as the primary factor opposing CBL
growth. Additionally, it is important to note that subsidence in complex terrain often results from horizontal convergence of
upslope flow branches detaching from the slopes (Schmidli, 2013; Adler and Kalthoff, 2014; Serafin et al., 2018), in addition
to large-scale weather patterns. Admittedly, subsidence on IOP 2b may have also been to some extent affected by the foehn.



Furthermore, although not synoptically generated, subsidence induced by mountain-plain-wind circulation penetrating deeper into the Alps during late afternoon could also have played a role in all IOPs (Lugauer and Winkler, 2005; Goger et al., 2019,

their Fig. 7). However, differentiating to what extent these additional factors contributed to subsidence during these four IOPs is out of the scope of the present study. We also emphasize once again our neglect of horizontal effects on $z_i$ tendency in Eq. 1, which we have not been able to infer from existing measurements.

### 3.2 Time-height evolution of spectrum width, potential temperature $\theta$, and CBL depth $z_i$

Figure 8 includes $R_B$-based $z_i$ estimates for the radiosondes launched between 06:00 to 15:00 UTC, whenever a $z_i$ estimate
was provided by the $R_B$ method, and also $z_i$ obtained using the bottom-up approach applied to spectrum width. The lack of data during early morning hours on IOPs 2a and 4 is due to overcast conditions (Fig. 8a, d). The sporadic data gaps below 200 m AVF are due to the dead zone of the lidar at Kolsass. Since the lidar at Hochhäuser was scanning with negative elevation towards Kolsass, spectrum width data available occasionally within this layer indicates valid data originating from the Hochhäuser RHI measurements.

In agreement with Fig. 4, $z_i$ estimates derived using the bottom-up approach on IOPs 2a and 2b showed decent agreement with those stemming from the $R_B$ method (Fig. 8a,b). The agreement between the bottom-up and $R_B$ methods holds well for IOP 3 too, however in case of the 11:00 and 13:00 UTC radiosondes on IOP 4, the $R_B$ method yielded systematically deeper CBLs. Further corroboration of the bottom-up approach is provided when considering the angle of the isentropes between 08:00 and 14:00 UTC on all IOPs, with more vertical isentropes present inside of the CBL compared to more slanted isentropes aloft.

Undoubtedly, IOP 3 stands out due to the growing influence of a deep, highly turbulent layer descending into the Inn Valley over the course of the day. This is due to the intensifying southerly to southwesterly foehn, associated with strong shear effects, near-neutrally stable atmosphere, and intense mountain wave breaking, leading even to rotors (Adler et al., 2021d, their Fig. 4). Although the descending foehn layer on IOP 3 did not only suppress the CBL growth, as indicated by the $R_B$-based $z_i$, it also limited the depth of the evolving upvalley flow, thus inhibiting the formation of a well-developed CVV found on otherwise
more quiescent IOPs 2a and 4 (B21).

     After roughly 14:00 UTC on IOP 2a (Fig. 8a), turbulence spread through the entire valley atmosphere, as indicated by the abrupt increase in spectrum width values of 0.5 a.u. on average. This corresponds to the presence of a CVV, which is a phenomenon generated by a force imbalance along the vertical axis of terms in the lateral momentum budget (B21). The shear character of the MoBL at this time is further supported by the presence of highest spectrum width values below 250
m AVF, in excess of 0.7 a.u., demonstrating the presence of a low-level upvalley jet (not shown), which is responsible for CVV occurrence. It is worth noting that the performance of the bottom-up method becomes inferior in such a shear-dominated environment, given that the spectrum width does not at all cross the threshold previously designed for convectively-driven situations only.

     Between 16:00 and 22:00 UTC on IOP 2a, for instance, during the time when the upvalley flow jet is decoupling from
the surface (B21, their Fig. 8), the bottom-up approach is once again able to retrieve the depth of primarily shear-generated turbulent zones. However, this layer no longer possesses any meaningful connection to buoyantly-driven turbulent layers, since





the sensible heat flux $H$ became negative several hours earlier (Fig. 7c). They are probably linked to the decay of the CVV,
which appears to be rather non-uniform with height.

### 3.3 Identifying distinct regimes of daytime MoBL spatial evolution

Previous conclusions made based on Figs. 7 and 8 suggested a complex evolution of the MoBL above the Inn Valley floor.
In the following we define three distinct stages of MoBL evolution, during three identical time windows (09:00 - 10:00 UTC,
12:00 - 13:00 UTC, 15:00 - 16:00 UTC, indicated by the vertical orange shaded regions in Fig. 7), now also considering the
spatial variability in the cross-valley transect. This characterization is once again enabled by the tight chronological succession
of the four IOPs, allowing us to neglect seasonality, and by extension, varying daytime duration. Additionally, we highlight the
fact that, although our treatment of the artificial oscillatory patterns in the RHI measurements from the Mairbach lidar were
successful when considering instantaneous, 1-min snapshots (Fig. 2), the act of computing hourly averages of spectrum width
teased out some remaining artefacts in the cross-valley transects (e.g. Fig. 9g,h,i,l). Since these situations occurred mostly
during the later part of day when the performance of the bottom-up method is compromised anyway, it does therefore not
affect our findings made for CBL states earlier during the IOPs.

Of all three time windows, the first one between 09:00 and 10:00 UTC (Fig. 9a-d) demonstrates the greatest degree of
similarity across the IOPs. At this time, the along-valley flow is relatively weaker than later in the day (Fig. 7a), the sensible
heat flux is at its most intense (Fig. 7c), while the CBL growth rates among the IOPs are fairly comparable (Fig. 7d). Mostly
terrain-following CBL top as well as an elevated spectrum width region fixed to the southern edge of the plateau denote the main
characteristics of this regime. The former characteristic is a well-known property of morning CBLs in valleys (De Wekker and
Kossmann, 2015), while the latter characteristic suggests highly turbulent upslope flows (Adler et al., 2021d, their Fig. 7c,d),
in the form of thermals, detaching from the plateau owing to their inertia. An instantaneous detachment of a thermal from the
plateau is evident in Fig. 3. Despite the developing foehn on IOP 3, at this point still limited to the ridgeline level, the CBL was
nonetheless able to develop in a manner similar to the other IOPs (Fig. 9c). As far as sidewall contrasts are concerned, CBLs
up to 200 m deeper above the plateau than over the southern sidewall are a common feature of all IOPs, with the exception of
IOP 2b.

By 12:00 UTC (Fig. 9e-h), significant deviations in the CBL development have arisen. At one hand, the upvalley flow is
on average gaining in intensity (Fig. 7a,b), while on the other hand the sensible heat flux resumes to weaken, changing sign
within the next hour (Fig. 7c). The cross-valley CBL structure among the IOPs has lost any degree of similarity present just
three hours ago (Fig. 9a-d). Of the four IOPs, only IOP 2a (Fig. 9e) has achieved one of possible textbook stages of CBL
development in a valley, namely a level CBL top independent from terrain (De Wekker and Kossmann, 2015). On IOP 2b (Fig.
9f), the CBL has slightly deepened or at least remained constant over the slopes compared to the morning regime, while it
became shallower over the valley floor, as shown already in Figs. 7d and 8b. The continually descending foehn layer on IOP 3
(Fig. 9g) has reached the undisturbed CBL by this time, resulting in the inability of the bottom-up method to yield meaningful
$z_i$ anymore. Interestingly, despite the overall similarity between IOPs 2a and 4 (Figs. 7, 8), the cross-valley CBL structure on
IOP 4 retained its terrain-following structure. Lastly, it is noteworthy to mention that the plateau-locked upslope flows were



still present at this time on all IOPs except IOP 2b, roughly with the same degree of turbulence intensities as three hours prior. Although buoyancy still represents a driving mechanism at this time, increased shear via intensifying upvalley flow as well as varying synoptic influences have led to pronounced differences among the IOPs.

With the upvalley flow having reached its pinnacle between 15:00 and 16:00 UTC (Fig. 7a,b), in the face of a negative
sensible heat flux (Fig. 7c), the well-mixed boundary layer across all four IOPs entered a primarily shear-driven regime (Fig. 9i-l). The cross-valley spectrum width fields are largely characterized by the inadequacy of the bottom-up method in identifying the CBL top $z_i$ on one hand, and by the uniform region of surface-attached high spectrum width, particularly on IOPs 2a and 4 (Fig. 9i,l). As noted earlier, this points to the low-level upvalley flow jet. Additionally, the elevated turbulence near the ridgeline level could also partly be due to penetrating mountain-plain-wind circulation at this time of day, as well as foehn in case of
IOPs 2b and 3 (Goger et al., 2019).

To summarize, despite having started from a state of textbook-like, terrain-following CBL structure during the morning (Fig. 9a-d), the CBL during each IOP experienced a substantially different structure by late afternoon (Fig. 9i-l), depending primarily on the temporal evolution of upvalley flow, upper-valley stability, presence of foehn, and tentative penetration of a mountain-plain-wind circulation. As a result, the structure of the well-mixed boundary layer in the early evening period differed
drastically among the four IOPs.

### 3.4  Cross-valley variability of MoBL evolution on IOP 2a

Highly spatio-temporally resolved fields of CBL depth $z_i$ are readily available from modelling tools such as idealized large eddy simulations (LES). A common visualization technique for analyzing such fields is a time-distance Hovmoeller-style diagram, as found for instance in Catalano and Moeng (2010), their Fig. 7, or in Schmidli (2013), his Fig. 1b. Obtaining such
detailed spatial information of $z_i$ evolution routinely in time from observations remains a substantial challenge to overcome. To our knowledge, we report $z_i$ for the first time from observations in such a Hovmoeller-style fashion (Fig. 10). This novelty is further augmented, by also considering the evolution of subsidence in both space and time, which together with the inclusion of $\Gamma$ and $H$ enables a unique glance into CBL evolution (Eq. 2). Note that the $z_i$ reported in Fig. 10 is now in meters above ground level (AGL). We will focus only on IOP 2a.

As before (Figs. 7d, 8), we applied the bottom-up method to 1-hr averages of spectrum width, while also sliding this 1-hr window forward in time by 10 minutes to enhance visual detail (Fig. 10a). Some extreme cases remain suspect, such as CBLs shallower than 100 m, although we emphasize that these are most likely constrained by the lowest available range gate as opposed to being linked to the actual depth of the layer being currently considered. These cases are isolated and shown in yellow. Before exploring the CBL development on IOP 2a, we highlight the fact that availabilities of $z_i$ and subsidence are
not necessarily perfectly co-located, given the fact that when CBLs are still fairly shallow, they may be within the region of near-parallel lidar beams in the coplanar-retrieved vertical velocity field (Fig. 3), thus yielding no subsidence: for instance, the region around 1000 m south of Kolsass between 06:00 and 10:00 UTC (Fig. 10a,b). Unlike $\Gamma$ which is only available from a single radiosonde launch site, some spatial information of $H$ is indeed available, by considering the three i-Box flux towers closest to the cross-valley transect (Fig. 10c). Separate analysis based on computing $\Gamma$ from the cross-valley aircraft flight legs



on IOPs 2b and 4 (not shown) revealed that $\Gamma$ remains nearly constant across the valley, particularly before the onset of the CVV (B21, their Fig. 9c,f).

As already demonstrated through Figs. 7 and 8, the time period between approximately 06:00 and 13:30 UTC on IOP 2a (Fig. 10) is most relevant for equating the elevations identified by the bottom-up method as $z_i$, due to positive $H$ at this time (Fig. 10c). During this period, the valley floor experienced the largest CBL growth amplitude, while the plateau experienced a

sudden growth by 09:30 UTC, remaining fairly stagnant afterwards. Nonetheless, the CBL development is highly variable in space, promoting the importance of such observations for validation of both idealized and real-world LES. From 12:00 UTC until the onset of the CVV around 14:00 UTC, the deepest CBL is found above the valley floor and the base of the southern slope. After 14:00 UTC, very deep turbulent layers are registered, extending beyond 1200 m AGL, as over the valley floor between 14:00 and 16:00 UTC, a sign of the presence of CVV at this time. By 19:00 UTC, the CVV has begun to decay and

the bottom-up method performs reasonably once again. After 20:00 UTC generally low spectrum width values throughout the valley atmosphere indicate ubiquitous stable MoBL formation along the cross-valley transect.

The entire transect is also characterized by subsidence values between -0.1 and -0.4 m s$^{-1}$ (Fig. 10b). After 14:00 UTC, the subsidence is more erratic and thus loses its physical meaning for $z_i$ evolution, given the CVV influence. Between 16:00 and 18:00 UTC above the southern slope, the updraught branch of the CVV is visible (B21, their Fig. 3). It may be tempting to

assign the deepest CBL above the plateau up to 12:00 UTC to the largest $H$ at the Eggen station near the edge of the plateau (Fig. 10c). Similarly, it seems plausible to assign the deepest CBLs above the valley floor and base of the southern slope to the now greatest $H$ at Hochhäuser between 12:00 and 15:00 UTC (Fig. 10c). However, such direct attributions may be misleading, given the limited spatial representativity of $H$ in complex terrain.

## 4   Conclusions

This study presented an investigation of the CBL and MoBL evolution in the CROSSINN investigation area, located to the east of the city of Innsbruck in the Inn Valley, Austria, over a single week period during August 2019. Given the on-going difficulty of adequately labeling any turbulent layer as either a CBL or a MoBL over complex terrain, we stuck with a more general labeling equal to CBL until the sign reversal of the surface sensible heat flux $H$ from positive to negative, when we switched to denoting turbulent layers as constituting a well-mixed boundary layer. Although the well-mixed boundary layer

was mostly comprised of shear-driven turbulence, labeling it as the MoBL itself would not have been justified, given our inability to reliably capture the entire impact of the Alps on the lower troposphere. During this week, four IOP events took place, specifically 2a, 2b, 3, and 4. A unique approach of studying the different CBL stages during the IOPs was possible due to turbulent Doppler spectrum width. Labeled as spectrum width in this study, it is a quantity carrying the signature of buoyantly and mechanically generated turbulence within the pulsed volume of a Leosphere Windcube WLS200s Doppler lidar. Three

such lidars were deployed in a cross-valley transect, enabling not only the derivation of a true vertical and one horizontal wind speed component in the cross-valley plain with the coplanar retrieval method, but also extending the areal coverage of spectrum width, up to 4 km horizontally and roughly 2 km vertically with a temporal resolution of 1 minute.





By utilizing in-situ aircraft and radiosonde measurements, we were able to develop a bottom-up threshold exceedance method which relies on the contrast of elevated spectrum widths within the CBL against lower levels found in the upper

valley atmosphere and even the free troposphere. We have only been able to adequately estimate the CBL depth $z_i$ when the surface sensible heat flux $H$ was positive. To calibrate the method for convective circumstances, the optimal threshold was chosen based on a comparison against the $z_i$ determined with the bulk-Richardson number approach applied to virtual potential temperature $\theta_v$ from the radiosondes. Additionally, validation of spectrum width features in space against rapid in-situ turbulence aircraft measurements further corroborated the promising capabilities of spectrum width for CBL evolution. Specif-

ically, depending on the aircraft's vertical position with respect to local $z_i$, we found decent agreement between spatio-temporal average values of vertical velocity variances $\langle w'^2 \rangle$ and kinematic heat and moisture fluxes $\langle w'\theta' \rangle$ and $\langle w'q' \rangle$ with theoretical expectations in cases whenever spectrum width was elevated. When spectrum width was generally lower, as is the case above the CBL, these quantities reflected different sign relationships and magnitudes, indicating either the entrainment zone or the free troposphere.

Through an examination of the temporal evolution of spectrum width above the valley floor, we established the primary spectrum width features at each developmental stage of the MoBL during the four IOPs. The deepest CBLs were found on IOPs 2a and 4, during which synoptic influence on the valley atmosphere was weakest. On the other hand, shallowest CBLs occurred on IOP 2b, which was primarily attributed to the strongest upper-level $\Gamma$ and partly to foehn influence. Compared to HHF terrain, where CBL typically achieves depths up to 1500 m or more (Stull, 1988), the CBL in the investigation area

was rarely deeper than 900 m. This is primarily due to strong static stability and much stronger subsidence above the CBL than that commonly found over HHF terrain. In afternoon hours, the presence of the CVV was expressed via spectrum widths much larger than those found earlier in the day within the CBL. Since the CVV is primarily a phenomenon characterized by pronounced shear in face of a negative $H$, the poorer performance of the bottom-up method reflected this state. The most unique aspect of this study was the cross-valley spatio-temporal evolution of both $z_i$ and $w_L$ in the form of a time-distance

diagram, a visualization technique that has up to now been mainly possible using large-eddy simulations for these parameters. Overall, relying on a technique that depends on turbulence properties helped highlight the limited representativeness of point measurements in complex terrain. Even under clear sky conditions, the already heterogeneous character of the MoBL found in our cross-valley transect may become substantially more complex under synoptic influence. In such situations, traditional ABL depth detection methods become inadequate, invoking the need for methods which better reflect turbulence features, such as

those found during CVV occurence.

The main take-away of the present study is the introduction of three distinct regimes that characterize the state of the MoBL in the Inn Valley from sunrise to shortly after sunset. During the first regime the CBL may be expressed as a rapidly growing CBL, but also as just a part of the overall MoBL. This growth is driven primarily by $H$, while its main opposing factor is $\Gamma$. The end of this regime is characterized by highest $H$ achieved during the day, as well as a distinct terrain-following $z_i$ across

the valley, regardless of $\Gamma$ or $w_L$ magnitude. The second regime describes the period when $H$ is decreasing simultaneously with an intensifying upvalley flow. Effects of $\Gamma$ and $w_L$ now become increasingly more influential, leading eventually to a case-dependent cross-valley pattern of $z_i$, ranging from horizontally level to still terrain-following. The sign change of $H$,





accompanied with the strongest upvalley flow during the day, marks the onset of the third and final regime. If the appropriate conditions are met, namely a sufficiently potent upvalley flow and negligible synoptic influence, the well-mixed boundary layer

at this time and in this particular cross-section of the Inn Valley will primarily be characterized by the CVV, which typically dissipates entirely a few hours after sunset, marking the beginning of stable MoBL formation.

Despite advancing our understanding of the MoBL structure in the Inn Valley owing to a novel sampling technique, a number of unresolved questions remain. These pertain mostly to the along-valley MoBL heterogeneity, an aspect addressable incompletely from aircraft measurements in our case. Given the high cost of conducting aircraft flights, reliance on remote

sensing approaches is superior in this regard. To gain a deeper insight into heterogeneity, we argue that the spectrum width methodology could be extended as well, by coupling multiple remote sensing lidar systems along the Inn Valley. Doing so would in turn enable estimates of the still elusive horizontal advection terms (Eq. 1), neglection of which is unjustified in complex terrain. On the other hand, although we have been able to provide highly resolved cross-valley transects of $z_i$ and $w_L$, our conclusions were nonetheless restricted to having just three discrete sites offering $H$. To truly explore the CBL growth

framework, similar highly resolved transects of $H$ are necessary, a demand potentially fulfilled with an array of carefully sited scintillometers (Ward, 2017). We are confident that the upcoming Multi-scale Transport and Exchange Processes in the Atmosphere over Mountains (TEAMx) programme and experiment (Serafin et al., 2020; Rotach et al., 2022) will offer the necessary means and resources to address these remaining challenges.

*Data availability.* The aircraft data can be retrieved from Adler et al. (2021a), the data from the WLS200s lidars can be found under Adler

et al. (2021b), while the data from the other KITcube instruments is located at Adler et al. (2021c). The i-Box data may be provided upon request.

**Appendix A:  Treatment of artificial spectrum width oscillatory patterns**

A regular occurrence in the RHI scans performed by the Mairbach WLS200s lidar was the presence of oscillations in the spectrum width with increasing distance from the origin (Fig. A1a), exhibiting a constant wavelength of approximately 400 m

as well as a varying amplitude with increasing distance from the Mairbach lidar (Fig. A1b). These artefacts were particularly evident in cases of low ambient turbulent levels, or equivalently, relatively low spectrum width values not exceeding 0.55 m s$^{-1}$. Highly turbulent regions in the valley atmosphere masked the presence of the oscillations. Despite our exhaustive internal efforts together with the lidar manufacturer, no successful tracing of the origin of these oscillatory artefacts has been accomplished (Leosphere, personal communication). Similar artefacts were also present in the case of the Hochhäuser lidar,

but to a far lesser extent. Therefore, we do not correct the Hochhäuser data for these effects.

To remove the oscillatory patterns from the Mairbach spectrum width, we first isolated the largest values by imposing a threshold level of 0.6 m s$^{-1}$. Through examination of several RHIs covering a range of different atmospheric conditions (not shown), we determined that any values beyond this threshold may severely impact the polynomial fitting used to detrend the





data distribution in Fig. A1b. These outlier values may be either due to physical phenomena or due to large distances from the origin at Mairbach. The evidence of the former can be found around 1300 m to the south of Mairbach, where the tall column of turbulence is attributed to spectrum width values reaching up to 1 m s$^{-1}$. On the other hand, the evidence for the latter can be found in case of the grid points also achieving such large spectrum width, but being located over the southern valley sidewall, more than 3500 m away from Mairbach. For our exploration of CBL development, the retaining of the former is highly desirable, while we found that the effect of the latter on the merged spectrum width field is typically not detrimental.

By applying a polynomial moving average with a window size of 20 points to all the spectrum width values not exceeding the threshold of 0.6 m s$^{-1}$, we have largely isolated the dominant 400-m wavelength of the oscillations while still retaining the smaller-scale perturbations (Fig. A1b). Prior to detrending the valid spectrum width values with this moving average, their mean over the scan is computed and added back to the now detrended spectrum width distribution. The resultant spectrum width field, shown in Fig. 2c, is now sufficiently devoid of any oscillatory artefacts.

Although this approach of removing the oscillations from instantaneous, 1-min RHI scans proved to be plausible, it is not entirely perfect. For instance, note the poor comparison of the fitted polynomial with the spectrum width distribution in the first two troughs, between 200 and 1000 m away from Mairbach (Fig. A1b). We found that such imperfections tend to get exacerbated with long-term temporal averaging, as evident for instance in Fig. 9g,h,i,l.

**Appendix B:  Addressing the spectrum width range discrepancies between the lidars**

Another artificial difference in the spectrum width between the three lidars was the range of values they sampled. On average and compared to Kolsass as the reference data set, the lidar at Hochhäuser systematically sampled the lowest (Fig. B1a), while the lidar at Mairbach systematically sampled the highest spectrum widths (Fig. B1b). To account for this inconsistency and bring the bulk of the distribution of each pair of RHIs as close as possible to the 1-to-1 line on a daily basis (Fig. B1), we computed the medians of both distributions for each day, followed by adding and subtracting the respective median to and from

all RHI scans, respectively for Hochhäuser and Mairbach. We found that daytime offsets were sufficiently appropriate for this, furthermore the consideration of all 74 daily offsets for the entire campaign revealed a clear systematic ordering relative to the 1-to-1 line (not shown). From one day to the next, the variability of the daily medians were reflective of the average turbulence intensities sampled on any given day by each of the lidars. Although we also tested hourly offsets, no notable advantages were found compared to the daily approach used here.

*Author contributions.*  Nevio Babić: conceptualization; data curation; formal analysis; investigation; methodology; supervision; visualization; writing – original draft. Bianca Adler: conceptualization; data curation; funding acquisition; project administration; supervision. Alexander Gohm: conceptualization; data curation; funding acquisition. Manuela Lehner: conceptualization; data curation; resources. Norbert Kalthoff: conceptualization; supervision.



*Competing interests.* The authors declare no competing interests.

*Acknowledgements.* The CROSSINN project was funded by the Deutsche Forschungsgemeinschaft (DFG, German Research Foundation) – 406279610. The DLR Cessna Grand Caravan 208B flights have been kindly financed by the Karlsruhe Institute of Technology (KIT). Part of University of Innsbruck's contribution was financed through the research project PIANO funded by the Austrian Science Fund (FWF) and the Weiss Science Foundation under Grant P29746-N32 (A. Gohm) and through project ASTER funded by the EGTC European Region Tyrol-South Tyrol-Trentino and the FWF under grant IPN 101-32 (M. Lehner). We thank Paul Ladstätter for providing the $R_B$-based $z_i$
estimates. The personal communications with Norman Wildmann and Ludovic Thobois are greatly appreciated.



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





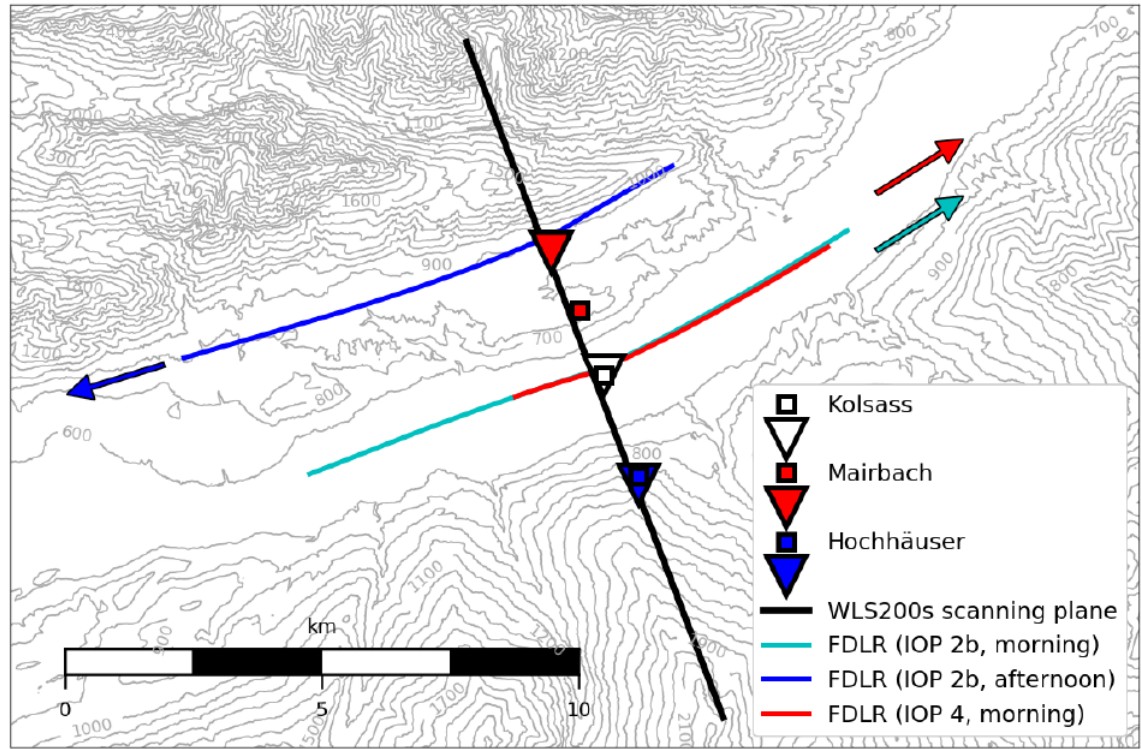

**Figure 1.** Overview map of the investigation area, centered around the Kolsass site at the Inn Valley floor. The square symbols indicate the locations of the permanent i-Box flux towers labeled with respect to their official designation (CS = core site; VF = valley floor; NF = north-facing; SF = south-facing; digit = local slope angle), while triangles denote the temporary KITcube WLS200s Doppler lidar locations. The solid black line represents the vertical plane of the coplanar-retrieved wind field. The three coloured solid lines represent the straight flight leg segments flown by the DLR Cessna aircraft during IOPs 2b and 4. Coloured arrows in the upper left corner depict the direction in which the DLR Cessna aircraft flew the respectively coloured flight legs.



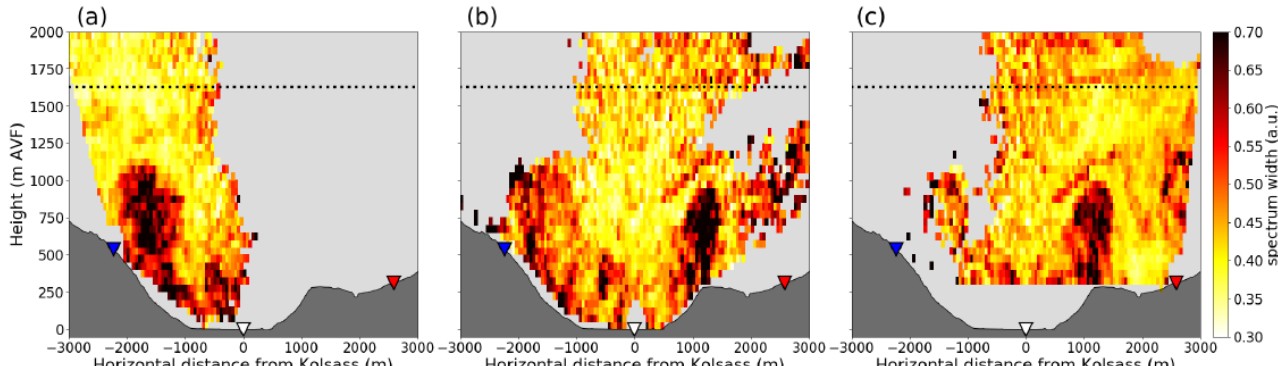

**Figure 2.** Instantaneous two-dimensional representation of the corrected spectrum width of the WLS200s lidar at **(a)** Hochhäuser, **(b)** Kolsass, and **(c)** Mairbach, valid for 10:58 UTC on IOP 4. Point of view is towards the southwest, i.e. in the upvalley direction. Coloured triangles denote the locations of the WLS200s lidars visible in Fig. 1. The horizontal dotted black line indicates the altitude of the average ridgeline level equal to 1630 m above valley floor (AVF).



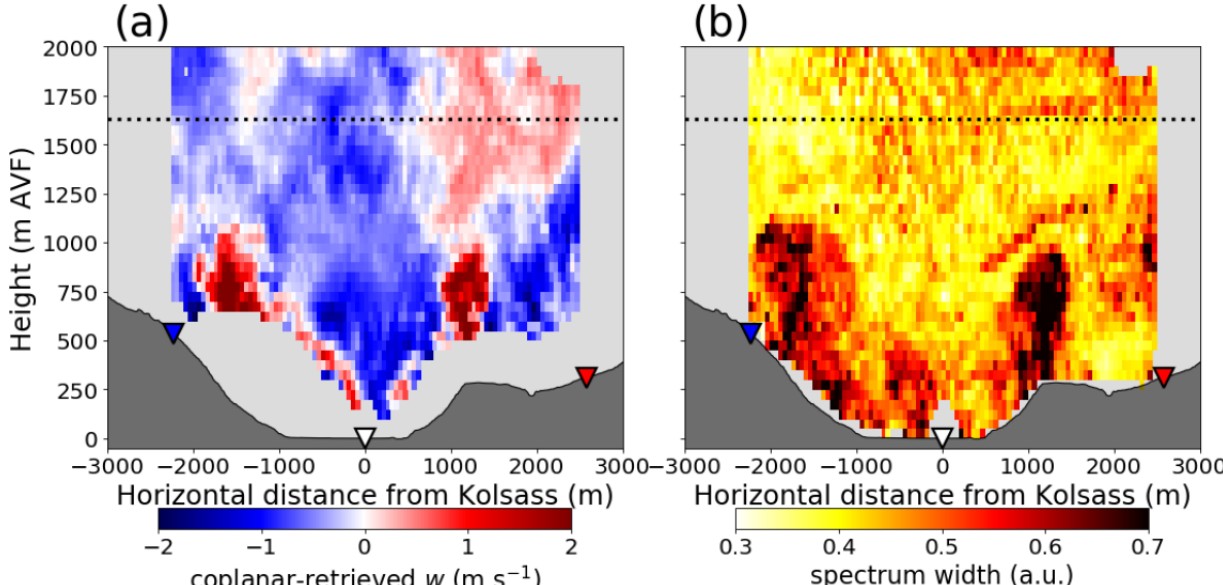

**Figure 3.** Instantaneous two-dimensional representation of **(a)** the coplanar-retrieved vertical velocity $w$, and **(b)** the averaged merged corrected spectrum width field from the three WLS200s lidars, valid for 10:58 UTC on IOP 4. Coloured triangles and the horizontal dotted black lines are as in Fig. 2.





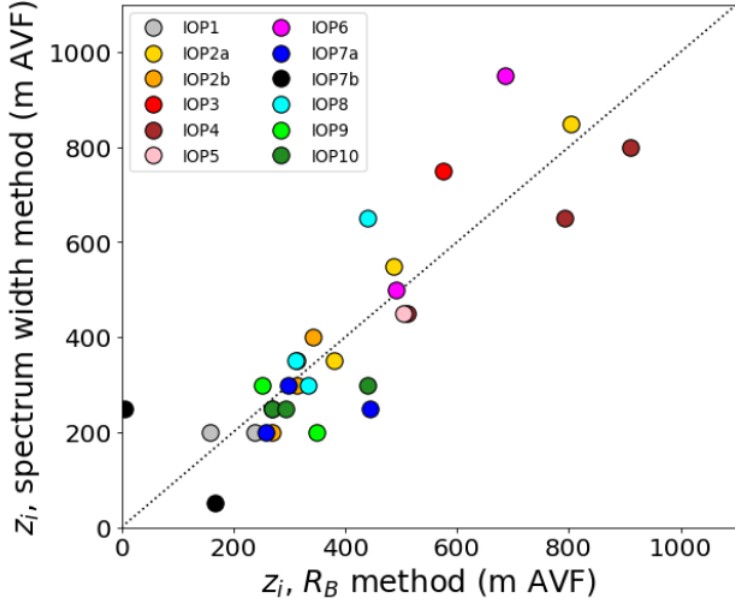

**Figure 4.** CBL depths $z_i$ obtained with the bottom-up method (threshold equal to 0.44 a.u.) applied to hourly composite merged spectrum width above Kolsass as a function of the $z_i$ obtained with the bulk-Richardson number method applied to the radiosonde virtual potential temperature (threshold equal to $R_B = 0.25$; $\theta_{v,0}$ calculated as the average of the lowest nine data points). Shown are only the radiosonde launch times at 9:00, 11:00, and 13:00 UTC, corresponding to the start of the 1-hour composite merged spectrum width mean. Markers are coloured according to their respective IOP. The thin dotted line represents the 1-to-1 line.



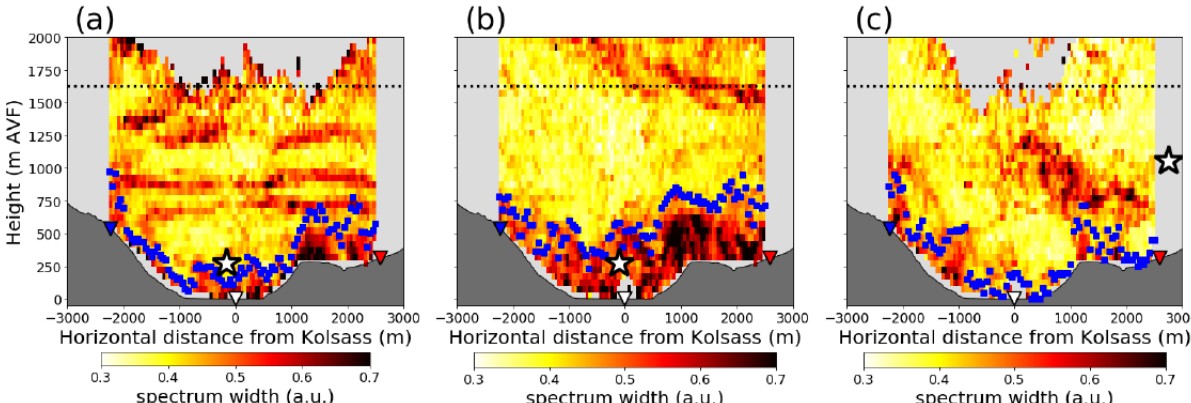

**Figure 5.** Instantaneous spectrum width fields during the times when the along-valley flight legs were flown closest to the WLS200s cross-valley transect, at **(a)** 08:26 UTC on IOP 2b, **(b)** 09:36 UTC on IOP 4, and **(c)** 14:20 UTC on IOP 2b. The blue squares represent the $z_i$ obtained by applying the bottom-up method to the 1-min corrected spectrum width fields, averaged across the duration of each straight flight leg segment. The white stars denote the flight leg position, while the coloured triangles and the horizontal dotted black lines are as in Fig. 2.



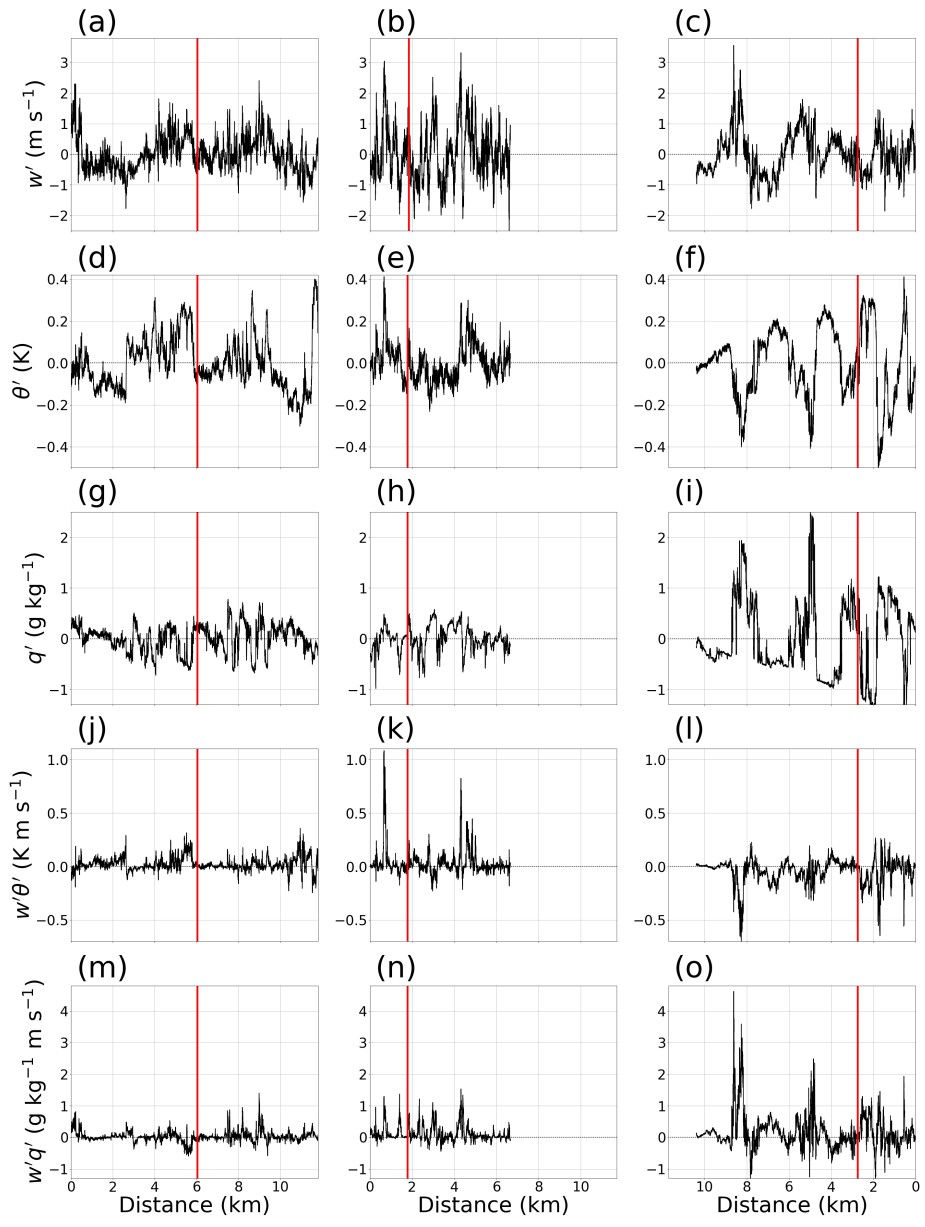

**Figure 6.** Along-valley, 100-Hz flight leg measurements made by the METPOD package aboard the DLR Cessna aircraft of **(a, b, c)** vertical velocity perturbations $w'$, **(d, e, f)** potential temperature perturbations $\theta'$, **(g, h, i)** specific humidity perturbations $q'$, **(j, k, l)** $w'\theta'$, and **(m, n, o)** $w'q'$. Shown are time periods corresponding to **(a, d, g, j, m)** 08:24:25-08:27:15 UTC on IOP 2b, **(b, e, h, k, n)** 09:35:40-09:37:25 UTC on IOP 4, and **(c, f, i, l, o)** 14:19:15-14:21:50 UTC on IOP 2b. The three flight legs were flown at average heights of 250, 250, and 1400 m AVF, respectively. The red lines denote the locations of the lidar cross-valley transect. The horizontal axis limits in **(b, e, h, k, n)** and **(c, f, i, l, o)** have been extended to match the longest leg segment in **(a, d, g, j, m)**. Note that the $x$-axis for the rightmost flight leg is reversed since this leg was flown in an opposite direction compared to the rest (Fig. 1), nonetheless the east-west orientation in each subplot shown here is universal.



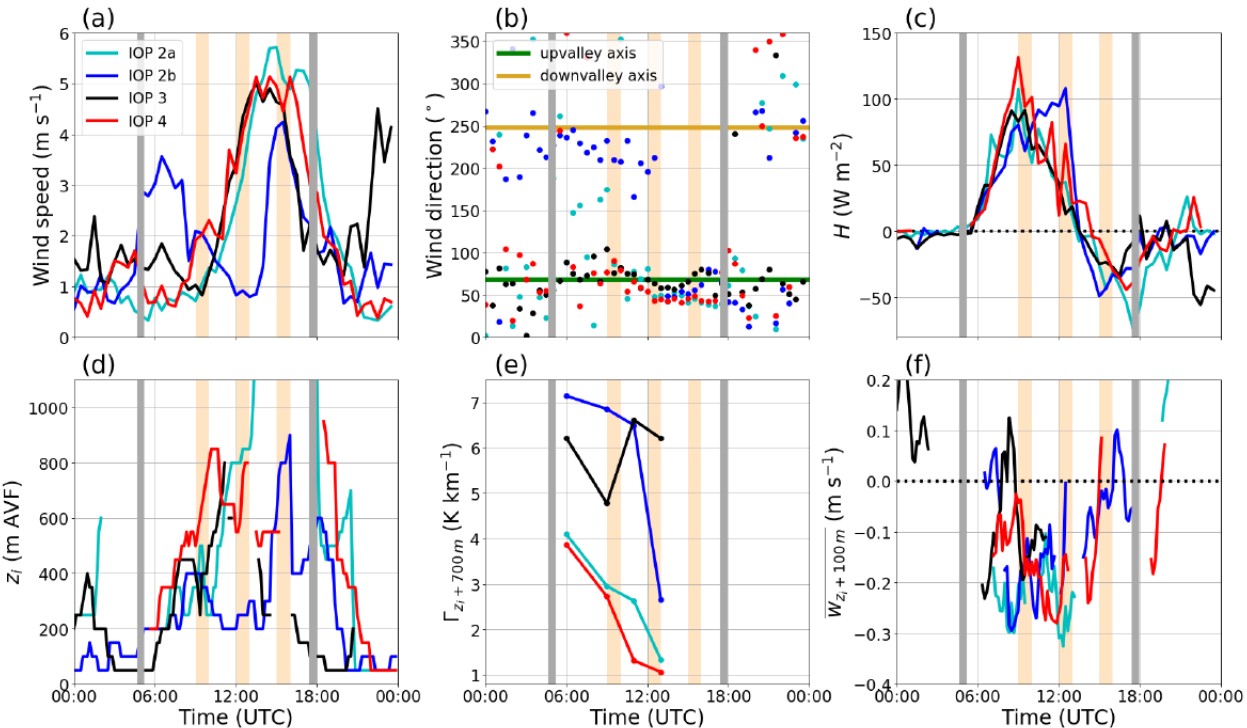

**Figure 7.** Time series of **(a)** wind speed at 8.7 m above ground level (AGL) at Kolsass, **(b)** wind direction at 8.7 m AGL at Kolsass, **(c)** sensible heat flux $H$ at 8.7 m AGL at Kolsass, **(d)** local $z_i$ above Kolsass, **(e)** mean lapse rate for the 700-m thick layer above local $z_i$ above Kolsass from the temperature $T$ calculated from radiosonde data, and **(f)** mean coplanar-retrieved vertical velocity $w$ within a 100-m thick layer above local $z_i$ above Kolsass, for the four IOPs shown in Fig. 8. The three orange shaded regions denote the respective 1-hr windows shown in Fig. 9. The two vertical grey bars denote respectively the local sunrise (04:54 UTC) and sunset (17:40 UTC) times on 11 August 2019.



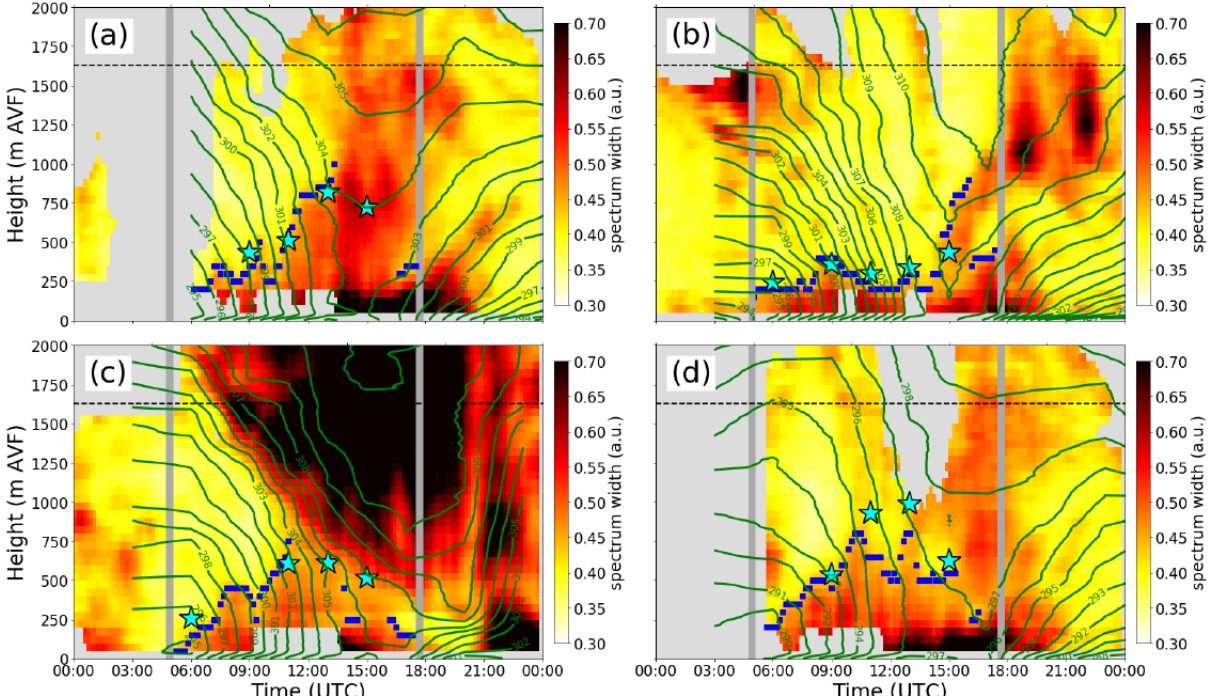

**Figure 8.** Time-height representation of spectrum width for the column above Kolsass (shading), radiosonde isentropes (green contours), $z_i$ obtained using the bottom-up threshold approach applied to spectrum width (blue squares), and $z_i$ obtained using the bulk-Richardson method (cyan stars), for **(a)** IOP 2a, **(b)** IOP 2b, **(c)** IOP 3, and **(d)** IOP 4. The $z_i$ obtained using the bottom-up threshold approach are omitted for nighttime periods, as the method was not calibrated to perform well during shallow stable boundary layer conditions. Each shaded spectrum width column is based on hourly windows shifted forward in time by 10 minutes for enhanced visual detail. The horizontal dashed black lines are as in Fig. 2, while the vertical grey bars are as in Fig. 7. To generate contour lines for the isentropes, all radiosonde launches from each IOP were used.




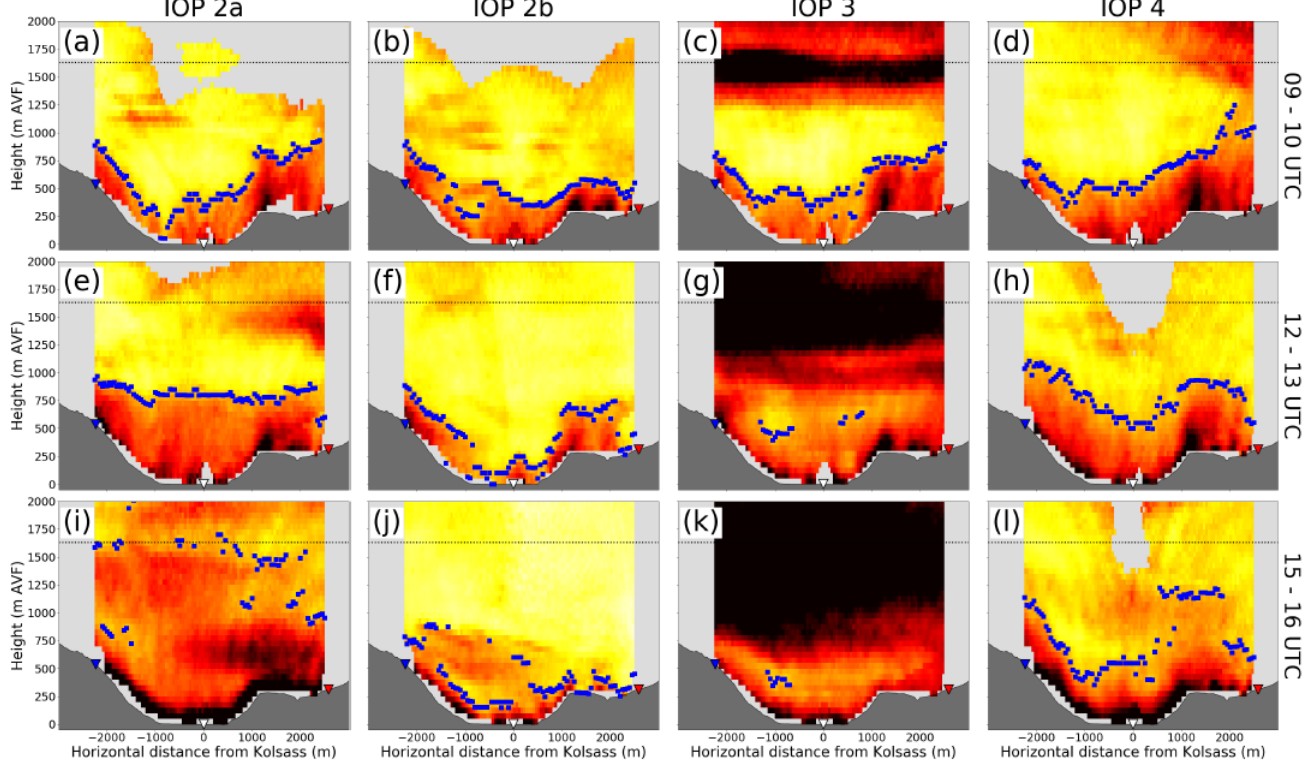

**Figure 9.** Cross-valley representations of 1-hr averaged spectrum width starting at **(a, b, c, d)** 9;00 UTC, **(e, f, g, h)** 12:00 UTC, and **(i, j, k, l)** 15:00 UTC, for **(a, e, i)** IOP 2a, **(b, f, j)** IOP 2b, **(c, g, k)** IOP 3, and **(d, h, l)** IOP 4. Blue squares denote $z_i$ obtained using the bottom-up threshold approach applied to each column of spectrum width. Coloured triangles and the horizontal dotted black lines are as in Fig. 2. The colorbar, omitted in favour of space, is identical to the ones found in Figs. 3, 5, 8.



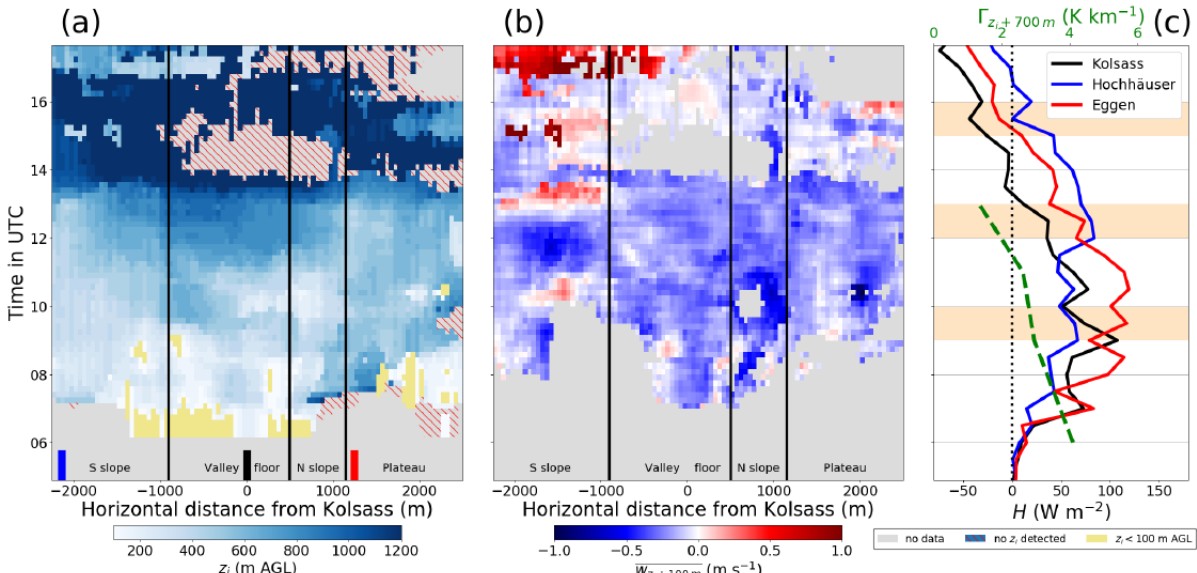

**Figure 10. (a)** Distance-time representation of $z_i$ (meters above ground level, AGL) obtained using the bottom-up threshold approach applied to each cross-valley column of spectrum width (shading), with additional indicators for no estimation possible (grey shading), for no $z_i$ detected (hatching) and for $z_i$ shallower than 100 m AGL (in yellow), **(b)** distance-time representation of the mean coplanar-retrieved vertical velocity $w$ within a 100-m thick layer above local $z_i$, and **(c)** time series of sensible heat flux $H$ from the three i-Box stations (in legend), together with the lapse rate for the 700-m thick layer above local $z_i$ above Kolsass from the radiosonde potential temperature $\theta$ (in green). Shown is IOP 2a. The vertical lines in **(a, b)** denote borders of the major terrain features in the valley cross-section (*S slope* = southern slope; *N slope* = northern slope), while the short coloured bars on the bottom correspond to the locations of the respective i-Box sites. The vertical axis has been constrained to the time between local sunrise and sunset.



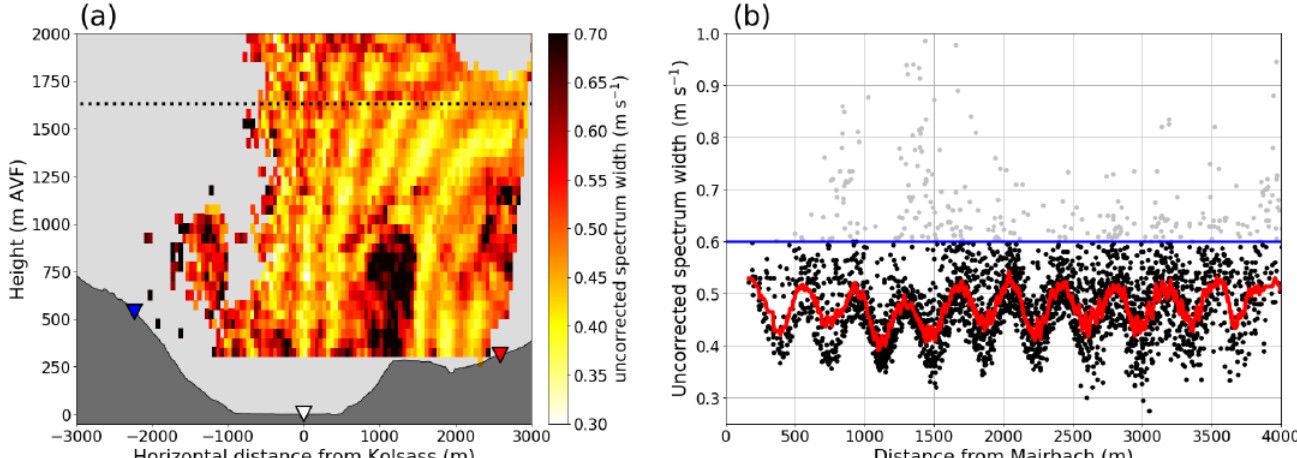

**Figure A1. (a)** Instantaneous two-dimensional representation of the uncorrected spectrum width from the perspective of the WLS200s lidar at Mairbach at 10:58 UTC on IOP 4. **(b)** Uncorrected spectrum width within each grid point of the coplanar retrieval mesh, plotted as a function of its respective distance from the WLS200s lidar at Mairbach, for the same time period as in **(a)**. Grey markers denote the grid points discarded owing to the threshold spectrum width (blue line), while the red line represents the moving average computed based on grid points not discarded by the threshold (black markers), equal to 20 grid points. Coloured triangles and the horizontal dotted black lines in **(a)** are as in Fig. 2.





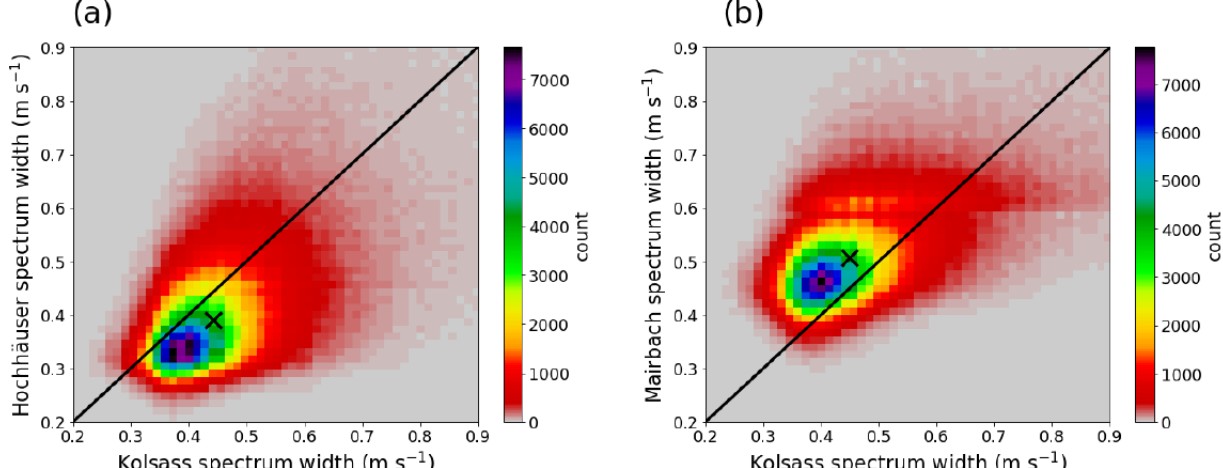

**Figure B1.** Heatmap representation of uncorrected spectrum widths from the **(a)** WLS200s lidar at Hochhäuser and **(b)** WLS200s lidar at Mairbach, plotted versus the uncorrected spectrum widths from the WLS200s lidar at Kolsass. Matching between the two RHIs is performed for each grid point of the coplanar-retrieval mesh and each 1-min case, for IOP 4 only. The diagonal solid black line represents the 1-to-1 line, while the black cross represents the median of the entire distribution.





|  | $\langle w'^2 \rangle$ (m$^2$ s$^{-2}$) | $\langle w'\theta' \rangle$ (K m s$^{-1}$) | $\langle w'q' \rangle$ (g kg$^{-1}$ m s$^{-1}$) |
|---|---|---|---|
| IOP 2b, morning | 0.31 | 0.02 | 0.03 |
| IOP 4, morning | 0.78 | 0.02 | 0.15 |
| IOP 2b, afternoon | 0.47 | -0.04 | 0.16 |

**Table 1.** Vertical velocity variance $\langle w'^2 \rangle$, kinematic sensible heat flux $\langle w'\theta' \rangle$, and kinematic latent heat flux $\langle w'q' \rangle$, for the three selected flights on IOPs 2b and 4 (Figs. 5, 6). Angle brackets indicate spatio-temporal averages computed as an average of the entire straight-leg segment flown along the valley. Turbulence perturbations are computed as deviations from a linear trendline.