# Peer review of "Exploring the daytime boundary layer evolution based on Doppler spectrum width from multiple coplanar wind lidars during CROSSINN"

_EGUsphere, 2023_

## Author Response (AR1)

**Response to the Reviewers' comments on the manuscript**
**Exploring the daytime boundary layer evolution based on Doppler spectrum width from multiple coplanar wind lidars during CROSSINN* [egusphere-2023-1977]**
**Babić et al**

We thank the two anonymous Reviewers for their constructive comments, critiques and suggestions, which have led to an improvement of the manuscript. Individual reviewer comments below are in bold font, while our responses are in regular font. Please note that, following the Copernicus guidelines for the write-up of final responses, we applied the *latexdiff* tool to the 1st and 2nd version of the manuscripts. The resulting PDF has been attached with this document. The lines we refer to throughout this response correspond to lines in this *latexdiff*-generated document, unless we refer to the Supplement (which is also provided with line numbering).

**Reviewer RC1**

**Summary: This study uses co-planar Doppler lidar observations in an across-valley transect during the CROSSINN campaign to (a) present a new approach for estimating the boundary layer depth and (b) applies these lidar observations along with other contextualizing field observations to describe variations in the MoBL structure and evolution. The new boundary layer depth approach relies on merged co-planar observations of the Doppler spectrum width, which is useful for evaluating CBL turbulence irrespective of the scan angle and thus overcomes limitation with other velocity based approaches that work for single column lidar observation but not RHIs. The analysis of the MoBL evolution reveals a mix of expected and unexpected features including terrain following CBLs, persistent thermalling features, and lowering Foehn flows, to name a few.**

**Overall comments:**

**The paper contains a number of useful and fascinating observations and techniques, but also suffers from a mix of focus on (a) a new technique and (b) trying to decipher MoBL physical processes. Broadly speaking I find the following strengths and weaknesses:**

We thank the Reviewer for these encouraging remarks.

**Strengths:**

- The use of the merged spectrum width to infer spatial variability in turbulent mixing is a strong contribution to the field of mountain boundary layer observations and alone warrants publication of this work. This approach overcomes the limitation of either using (a) single vertical velocity variance profiles above a fixed site, or (b) requiring dual-Doppler retrievals of the vertical wind, which can not yield the same spatially continuous coverage of the valley atmosphere.

- The paper clearly demonstrates that these merged spectrum width data reveal important spatially and temporally varying aspects of the MoBL that are not otherwise possible. The content of the individual figures is stimulating and informative, as is the analysis of them, which gets at many of the complexities of the MoBL as compared to its flat terrain counter part.

We thank the Reviewer for these encouraging remarks.

Weaknesses:

- The bottom up thresholding approach clearly has limitations, many evidenced in the analyses presented and some discussed by the authors. This is especially true in the presence of elevated turbulence sources. At times the approach seems to work great (e.g., Fig. 9a-f, h) whereas at other critical times it does not (e.g., Fig. 9g). Other examples of it not working well include the jumps in Zi apparent in Fig. 5a where the algorithm couples to the elevated layers aloft. This is not to say the approach isn't useful, it just adds another approach that suffers from many of the problems we always encounter in defining Zi.

We thank the Reviewer for this critique of the bottom-up thresholding approach. Indeed, in a couple of instances during the four chosen IOPs the algorithm appears to erroneously detect a height which is not necessarily linked to the CBL. We refer the Reviewer to later comments (also those by Reviewer #2), where we address some of those instances in more detail. However, we are confident that overall the algorithm performs well, especially in such a challenging complex terrain setting of the Inn Valley. Also, as the Reviewer indicates, the bottom-up method provides yet another useful tool to study the spatio-temporal behavior of the CBL/MoBL evolution.

- The approach is not all that well "validated" or "calibrated". Most of what you present are fairly loose comparisons with other imperfect observations (e.g., radiosondes, Rb, etc). This lack of rigor is somewhat unavoidable, but undermines the efficacy of the paper as a demonstration of technique in that the results are far from universal.

Although we understand the Reviewer's concern regarding an incomplete validation/calibration, we have to emphasize that these calibration/validation statistics were obtained based on traditional

in-situ measurements, namely radiosonde and aircraft. Both of these instrument platforms have been held as the *golden standard*, when it comes to ABL depth determination, for more than fifty years. We are aware of the limited sample size available for validation in our case. In other words, our validation approach is limited in the sense that it is performed for fair-weather summer days only, while its tentative usefulness for other conditions (e.g. when $H$ becomes negative) remains to be shown and quantified.

To help alleviate some of this otherwise totally justified concern, we show another *fairly loose* validation of the bottom-up method, this time against vertical velocity variance of the coplanar-retrieved vertical velocity $\sigma_w^2$ for a selected 30-min period on IOP 4 (one of the most vigorous convective periods during CROSSINN). The bottom-up-based estimates agree reasonably well with these additional (more traditional) variance-based estimates, further adding to the *validation* argument. However, in this case, two completely different spectral ranges of turbulence are involved (which do not necessarily interfere). Therefore, only qualitative comparisons of major CBL features are plausible. At this point, we decided not to include this sort of a comparison into the 2nd version of the manuscript, since we would introduce yet another *tool from the toolbox* into the mix (the $\sigma_w$ threshold) - which also has to be calibrated with respect to specific thresholds.

[Figure]

**Fig. 1: left** Vertical velocity variance of the coplanar-retrieved vertical velocity $\sigma_w^2$ and the $z_i$ obtained with a simple threshold exceedance method (cyan markers), and **right** $z_i$ (black markers) obtained with the bottom-up method applied to post-processed and merged spectrum width fields, for the period 10:30-11:00 UTC on IOP 4 (14 Aug 2019).

- **The paper is long and somewhat sprawling trying to accomplish two things at once (a) a technique demonstration and (b) a process-focused study. It leaves me wondering if splitting it into a technique paper and a process paper would be best. I'll defer to the authors and input from other reviewers on this, as I understand the motivation to merge the two. I ran a quick word count on this, and I think (could be wrong) that you're well over 10,000 words (probably 11-12,000). Many Journals try to keep word limits to ca 7500. In reading the**

> **text I estimate you could probably remove up to 25 % of the total words and significantly improve the readability and conciseness of your manuscript.**

We agree with the Reviewer that the original submission was somewhat long and designed in a way that shifted the focus from the process-based aspects, i.e. those that describe the thermodynamical evolution of the CBL and MoBL in this section of the Inn Valley.

However, due to time constraints (all the co-authors have moved onto other projects since CROSSINN), as well as career changes (the corresponding author left academia and is now working in aviation meteorology), we kindly ask the Reviewers to understand our desire not to split the current manuscript into two separate ones, or to move to another journal altogether. Therefore, we have asked both the co-editor and the Copernicus editorial staff whether it would be possible to move the bulk of the Methodology (sub)sections and both of the former Appendices to a separate Supplement document, which they have approved around the end of November. Similar was already done for our prior QJRMS cross-valley vortex paper (Babić et al, 2021), as well as for one of the PIANO papers (Haid et al, 2020).

- **Can you please specify what the units of the spectrum width are? I was assuming this was the Doppler spectrum width, thus representing the ranger of Doppler velocities sampled inside a lidar volume (pencil beam). It looks like your units are a.u.? I apologize for my ignorance, but I'm not familiar with what the a.u. unit is with respect to measured spectrum width. I'd have thought the units would be m/s?**

The Reviewer is indeed correct, the original spectrum width units are in meters per second. However, owing to multiple post-processing steps, on line 194 of the first version of the manuscript we already specified that a.u. refers to *arbitrary units*. The unit clarification has now been added to both the main text (caption of Fig. 2) and to the Supplement.

- **Lines 173-180: I don't understand why shear effects are separate from the true or measured value? How are they separable here?**

In our case, they are not separable, as we do not calculate any of the individual contributors to $\sigma_t^2$. Instead, we rely on the spectrum width (broadening) product output by the Leosphere software Windforge. We presented the relevant equation in the 1st version (which remains in the 2nd version too) only as an introduction to the concept of spectrum width (broadening).

- **Line 183: A fundamental question: if the spectrum width is related the range of observed Doppler velocities in a sample time/volume then are there expected to be differences as a function of elevation angle in an RHI for non-isotropic**

**turbulence? In other words if the flow is a convective CBL with positively skewed vertical velocity you probably get a rather different spectrum width for the near horizontal vs near vertical parts of the RHI. Has this been explored? Is it an important consideration in merging the RHI scans?**

It is correct that, within a highly anisotropic CBL flow, the spectrum width may vary depending on how the laser beam (at a certain elevation of the RHI scan) intersects the flow. Smalikho et al (2005), in their mathematical considerations of calculating each of the terms that contribute to $\sigma_t^2$, assumed purely isotropic turbulence. Prior to their Conclusion section, however, they do acknowledge that applying their isotropic-based model may be inadequate when the flow is anisotropic and/or inhomogeneous, since different integral scales of turbulence are involved in that case.

Ultimately, this omission of accounting the effect of RHI intersecting different thermals at different angles does not impact our work, since our instrument setup samples the valley atmosphere from essentially three different angles from below. Although the magnitude of spectrum width might still be affected, we hypothesize that this discrepancy is still much smaller than the magnitude difference of spectrum width between CBL/MoBL and free tropospheric/low-turbulence air aloft - as this contrast is exactly what we rely on in this study. To cover the case of a reader postulating a similar question, we have added this disclosure in the new Supplement document on line 91.

- **Section 2.4: I would recommend changing the title of this subsection to something more like "comparing spectrum width and vertical velocity". I was expecting some sort of statistical validation, whereas you provide a qualitative comparison (which is certainly still useful and worthwhile).**

We have changed the title of this subsection to *Comparing features in spectrum width and vertical velocity* on line 265.

- **Line 233: I don't understand the meaning of "1 hour intervals shifted forward by 10 minutes can you clarify"? Does this mean you take hour long averages every 10 minutes?**

That is correct. To help the reader with interpreting this, we have added an example of this in the Supplement on line 113.

- **Line 235: What is a.u.? Here and elsewhere.**

We refer the Reviewer to their previous comment where this has been addressed.

- **Line 242: errant apostrophe?**

That is correct. The unnecessary apostrophe has been removed in the revised version on line 94.

- **Line 266: This threshold (1100 m) gives me pause. I fully appreciate that these definitions always require range checks and thresholds but 1100 m seems well within the range of potential CBL depths in the MoBL... this approach discards any possibly deeper CBLs and preclude the application of the technique across a broader data set or other locations. I think you're safe here in that your results are reproducible for this particular study, but it will undermines broadness of the use of this technique.**

We thank the Reviewer for this remark. Indeed, the bottom-up method, calibrated specifically for this study, might not be extendable with this threshold depth of 1100 m to other valleys. In other words, it would have to be set separately for another valley, which we do not necessarily consider to be something that undermines the broadness of the use of this method.

The motivation behind the apparently *too shallow* threshold of 1100 m AVF in our case is twofold:

- This threshold, based on the validation of bottom-up-based $z_i$ estimates against those from the bulk-Richardson number $R_B$ approach is encapsulating the range of $z_i$ encountered during CROSSINN and for all ten considered IOPs. In more colloquial terms, this threshold *covers all the bases* when we consider 1-hr $z_i$ averages, as in our case;

- Indeed, there may be instances when the instantaneous, 1-min $z_i$ exceeds this threshold. One such case, corresponding to a thermal that occurred on IOP 4, depicts the deepest thermal sampled during these four IOPs (Fig. 2), reaching a depth of approximately 1300 m slightly to the southeast of Kolsass. Since this thermal and its advection by the upvalley flow through the RHI plane have ultimately been accounted for in the calculation of the 1-hr average (which amounted to 850 m AVF), we consider this threshold to be overall sufficiently well suited for the purpose of our study.

[Figure]

**Fig. 2:** Instantaneous two-dimensional representation of **left** the coplanar-retrieved vertical velocity $w$, and **right** the averaged merged corrected spectrum width field from the three WLS200s lidars, valid for 10:52 UTC on IOP 4.

- **4. These data look promising, but it would be interesting to know how much better or worse the RMSE values were for other combinations? For example was the 12 point theta average not wildly different, or pretty similar? What about 3 points?**

According to Fig. 3, the lowest $RMSE$ equal to 109 m, interpretable as the tightest scatter around the 1-to-1 line in Fig. S4 of the new Supplement document, is achieved with a combination of 0.44 a.u. threshold and a 9-point surface $\theta_{v,0}$ average. This figure has now been inserted into the manuscript, specifically into the new Supplement document on line 150.

[Figure]

**Fig. 3:** Density heatmap demonstrating the variation of the root mean squared error $RMSE$ of the $R_B$-based calibration of bottom-up-based $z_i$ estimates. Spectrum width thresholds range from 0.35 to 0.47 a.u., while surface $\theta_{v,0}$ average point number ranges from a 3-point to a 12-point average.

- **5a. In panel a, your CBL height ID on the left edge ( -2200 m) jumps up to the elevated spectrum width layers. Is this physical or not? Likewise at 2100 m (to the right) the blue dots jump to the elevated feature despite passing through a minima is spectrum width above the shallower near surface high SW layer. How do you interpret this result? I'm aware how tricky these threshold approaches are, and am supportive of the overall approach, it just seems there are some difficulties in its universal application (as with other approaches, the problem is not unique to you).**

We agree with the Reviewer that the application of a method, with a single threshold as in our case, cannot be expected to universally distinguish between surface-based, convectively-driven turbulence and elevated turbulence layers. Given the broad range of turbulence sources in a mountain valley, we initially determined that introducing *too many* thresholds (with the intent of covering all possible scenarios) would have made analyses overly complicated and even less representative for other complex terrain settings. This is also the reason why we dedicated a bit more focus to the valley floor in the manuscript, as opposed to making similar analysis for the more challenging slope regions.

- **Line 316: I'm not sure that a "mature CBL" is a defined concept, and if it were I probably wouldn't associate it with being decoupled, but rather strongly coupled. Consider rewording or specifying what you mean by mature?**

In this case, under *mature* we implied the state of the CBL when the surface sensible heat flux $H$ decreases and becomes negative, while a residual layer may begin to develop at higher altitudes. This is made even further complicated in our case, where the intense upvalley flows may advect deep thermals and/or their remnants from upstream regions (where $H$ may have already changed sign in the interim). To help reduce some of the confusion the Reviewer encountered, we modified the first sentence of the relevant paragraph on line 353.

- **Line 318: what allows you to characterize this elevated turbulence as convectively driven. Its decoupled nature would seem to suggest it has a non-surface based origin. Is it possibly mechanically generated? How do you know?**

Going back to the previous comment by the Reviewer: yes, this turbulence might also be mechanically driven indeed. To decrease ambiguity or making overreaching statements like this in the first version of the manuscript, we have omitted *convectively-driven* on line 355.

- **Line 325: These seem like very broad features to be convectively driven (1.5 km in width). Are they perhaps wave driven? Or waves coupled to the CBL top?**

Given the strong spatial co-location of major features in $w'$, $q'$ and $\theta'$ perturbations, we argue that these features are not a signature of a (gravity) wave. If that were the case, $w'$ and $\theta'$ would be 90 degrees spatially out of phase, i.e. shifted horizontally (which is not the case here).

- **Line 333-334: But the cross section doesn't show any coupling to the surface, so it is either not representative of the valley as a whole, or these are a not coupled to the surface? I'm a bit confused here.**

In that last sentence of the paragraph, we did not state that the observed thermals were coupled to the surface, just that similar thermal signatures were observed across several kilometers above the plateau.

- **8. We can again see some of the issues with the bottom-up threshold approach where your algorithm seems to jump to elevated layers (e.g., Fig. 8b at ~1600-1700**

UTC), or wherein there are strong elevated sources of spectrum width not driven by CBL processes (e.g. strong winds aloft). This also gets at the issue of the 1100 m height threshold imposed. . . there seems to be coupling processes at play that *could* produce deeper convectively (or shear and convective) boundary layer depths).

We once again agree with the Reviewer (see the earlier comment and our reply above). We now realize that the presence of these mis-identifications may introduce confusion, therefore we have added the following *disclosure* on line 502 to improve the understanding of what processes the algorithm is specifically designed to target.

- **Line 468: This is a really interesting persistent thermal plume feature. Really underscore how nice the merged RHI data are for examining the MoBL.**

Agreed. Of course, the thermals at the edge of the plateau do indeed *come and go* at varying stages of their development. However, their persistent signal in the 1-hr average across all IOPs corroborates that this feature is indeed tied to the terrain.

- **9. This figure is fantastic and provides all sorts of interesting details about the structure of the MoBL, including both terrain following and non-terrain following components of the MoBL structure, cross valley asymmetries (e.g., Fig. 9d).**

We thank the Reviewer for this encouraging remark.

- **Line 497: I'm a bit confused by the surface following high spectrum width feature and your description. First, just to be clear, is this evidence of a near surface up or down valley flow? Second, this feature is so strongly terrain following I'm almost confused by it. What does a radiosonde wind profile look like at this time? Does it show a low-level jet feature near the surface?**

In this case (fair-weather *quiescent* IOPs when thermally-driven flows develop unperturbed for the most part), the surface flow is upvalley. In fact, as seen in Fig. 4 which is taken from B21, it is visible that the entire valley atmosphere below the average ridgeline level is characterized by this northeasterly, upvalley flow. Furthermore, the upvalley flow jet profile can also be seen from Fig. 5 (also found in B21). It is the portion of this jet below the maximum and all the way down to the surface that is responsible for the extremely high values of spectrum width found there since turbulence shear production, owing to large values of $dU/dz$ (and possibly also of momentum covariance $\overline{u'w'}$), is significant at this time of day.

[Figure]

**Fig. 4:** Cross-valley representation of the FDLR flight legs on IOP 4 (2nd afternoon flight), while black contours represent the along-valley wind speed component in the natural frame of reference (positive/solid = upvalley; negative/dashed = downvalley). Going from white to red shading signified an increase in horizontal wind speed magnitude. The horizontal dotted black lines indicate the altitude of the average ridgeline level equal to 1,630m AVF. *(Taken and adapted from B21, their Fig. 9)*

[Figure]

**Fig. 5:** Vertical profiles of Halo Photonics SLXR142 lidar velocity-azimuth display (VAD) (a) wind speed and (b) wind direction, for the matching 30-min intervals with a CVV. The horizontal dotted black line indicates the altitude of the average ridgeline level equal to 1,630m AVF. *(Taken and adapted from B21, their Fig. 6)*

- **Lines 540-546: Seems odd to start a conclusions with a bunch of caveats. I'd recommend revising this section and focusing on what you have established. The rest of the paper addresses the nuances.**

We have reworded this beginning part of the Conclusion (on line 614) to better highlight the advantage we get from setting specific labeling into a CBL and a well-mixed layer. Although we

did not initially consider our first approach as necessarily listing out caveats, we agree with the Reviewer that emphasizing more clearly the target and advantages of our approach at the start of the Conclusion is overall a better way to go.

**Reviewer RC2**

This manuscript introduces a novel method to retrieve the convective boundary layer (CBL) height from coplanar Lidar scans. The scan data is availbale from the CROSSINN campaign, which took place in an Alpine Valley to study, among other mountain boundary layer (MoBL) phenomena, the cross-valley wind circulation. The new method to determine the CBL height is described over around 8 pages of the manuscript (excluding Figures) and gives a detailed overview of the necessary assumptions to extract the CBL height information from the measurements, which can be considered as a first part. In the second part (Section 3 onwards), the authors describe the varibility over space and time of selected ABL variables influencing the CBL height. Finally, they identify distinct regimes of MoBL (spatial) evolution determined from four selected intensive observation periods (IOPs). This work substantially contributes to the rich body of boundary-layer research in the Inn Valley and highlights the complexity of mountain boundary layers (again). Unfortunately, the manuscript reads more like a measurement report than a paper manuscript, because the authors leave many questions open, while they could have answered them by extracting more information from the CROSSINN and i-Box datasets.

We thank the reviewer for their remarks.

- The authors try to bridge the gap between introducing a new method to determine the CBL height by analyzing the measurement data to gain new knowledge on the boundary-layer evolution in the Inn Valley, Austria. Since the explanation of the method is complex, this already takes almost half of the entire manuscript, which is very long, as a previous referee already pointed out. The extensive description of a measurement setup is likely out of scope for Weather and Climate Dynamics: The detailed description and proof of concept likely fit better in other Copernicus journals as Geoscientific Instrumentation, Methods and Data Systems (GI) or Atmospheric Measurement Techniques (AMT). At AMT, there is even a well-fitting special issue open right now: "Profiling the atmospheric boundary layer at a European scale" (link). Please consider a split of the manuscript, and then you can focus in the current WCD manuscript almost solely on the boundary-layer dynamics part. Henceforth, the major part of my review will focus on the interpretation and results from Section 3 and onwards.

We thank the Reviewer for their critique of the style of the first version of the manuscript. We wish to point the Reviewer to a directly related comment from the first Reviewer a few pages above where our justification for avoiding splitting and/or moving the manuscript elsewhere has already been laid out. We reiterate that response once again here:

Due to time constraints (all the co-authors have moved onto other projects since CROSSINN),

as well as career changes (the corresponding author left academia and is now working in aviation meteorology), we kindly ask the Reviewers to understand our desire not to split the current manuscript into two separate ones, or to move to another journal altogether. Therefore, we have asked both the co-editor and the Copernicus editorial staff whether it would be possible to move the bulk of the Methodology (sub)sections and both of the former Appendices to a separate Supplement document, which they have approved around the end of November. Similar was already done for our prior QJRMS cross-valley vortex paper (Babić et al, 2021), as well as for one of the PIANO papers (Haid et al, 2020).

- **The second part is written in an almost chaotic way. The authors assume that the reader already knows a lot about (i) the Inn Valley and surroundings, (ii) the local boundary layer development, and (iii) read all the previous publications of the CROSSINN campaign (e.g., different IOPs are mentioned, but there is no description on which processes were actually at play or that actually happened in the IOPs). This makes it difficulat to understand the relevant phenomena at play.**

We agree that insufficient attention was dedicated to a complete description of processes taking place in the Inn Valley. This has been significantly expanded upon in the second version of the manuscript, based on the Reviewer's suggestions that follow.

- **Furthermore, the authors leave a lot of questions open. On the one hand, they could answer them easily instead of speculating (e.g., using observations of radiosondes or the eddy covariance towers, or aircraft data), and on the other hand, they could extend their comparison to more IOPs (e.g., in Section 3.4) or add a fifth IOP with less synoptic influence (e.g., IOP10). If the authors choose to split their manuscript, this can be easily achieved.**

Although we understand the Reviewer's reasoning concerning the benefit of extracting more interpretation from point measurements (i-Box) or single vertical profiles (radiosondes), we argue that extending the analysis beyond what has already been presented could represent a sort of a *double-edged sword*. By this we mean that *over-interpreting* beyond the analyses presented thus far, the e.g. highly locally limited i-Box data sets might give an incorrect image of the actual behavior of flows in the area of interest. Furthermore, by adding more figures into the now split-up second version of the main text, we would once again begin running into the issue of exceeding a tolerable word count, as the first Reviewer has already pointed out above.

On the other hand, we argue that adding an IOP (specifically, IOP 10) that is so far temporally apart from the four already considered ones, would add even more complexity in the interpretation of the results. Since daylight duration was already significantly shorter by the time IOP

10 took place, energy partitioning at the surface would reflect this, thereby resulting in shortening/lengthening of the three regimes found to be valid for the four August IOPs. The *constrained seasonality and tight occurrence* argument for choosing the four August IOPs would thus be weakened. Finally, although IOP 10 may appear to have been a quiescent IOP, it was marked (using the Plavcan et al, 2014 foehn probability algorithm) as an IOP with high foehn probability. Therefore, we did not include any additional IOPs into consideration in the revised 2nd version.

- **Comparison with other work and lack of discussion - the comparison with real-case and idealized simulations is of course valid, especially when there is a lack of previous observations. However, I wonder why the authors complexely omit a comparison with other measurement campaigns, e.g., Hymex (Adler and Kalthoff, 2014, 2016); MAP-Rivera (Rotach et al, 2007, and their previous papers on the topic), T-REX observations and simulations (Strauss et al, 2015; Babic et al, 2019), and PIANO on foehn flows (Haid et al, 2020, 2021) to put their results into context.**

During the initial writing stage, we pondered about having a separate Discussion section, where certain aspects of the study thus far could be put into a broader context. At the time, space limitations were the reason for not having such a section. Now, while we still decided not to have a separate Discussion section, we have, as each Section progresses and where necessary, already there made comparisons and citations to past work over mountain valleys - particularly in *Section 2: Typical evolution of the MoBL in the investigation area during August.* Also, initially we found that making such comparisons would, to a certain degree, be analogous to *comparing apples and oranges* - meaning that the high specificity of other valleys or measurement campaigns prevented any reasonable comparison to begin with. However, we realize now that making such comparisons may only strenghten our conclusions regarding our own findings.

**To make the manuscript fit within the scope of WCD, the following possible questions could be answered:**
We greatly appreciate the Reviewer's efforts in helping us sharpen the manuscript's storyline. We have integrated their five recommendations in the following manner:

- **Is there such a thing as an "ideal CBL development" in complex terrain?**
- **(already partly answered) Which processes lead to a non-ideal CBL development? Typical MoBL processes due to the underlying terrain such as up-valley winds, slope flows, and the plain-to-mountain circulation, or other, larger scale influence (e.g., chanelling, foehn flows)?**

As the Reviewer already hinted, throughout the text and by analyzing the results, we have already

partly addressed these two related questions. In line with the Reviewer's previous comment about citing past work and literature, we have bundled these questions with the earlier *fortification* of our conclusions via citing more relevant studies.

- **Does the new CBL height determination method help us to untangle this complex flow structure, or does it raise new open questions? Is a regime classification with schematic diagrams possible, as in, e.g., Haid et al (2022)?**

We expanded and adapted the last paragraph of the Conclusion section on line 664, describing the new questions raised by the application of the bottom-up exceedance method. Hopefully these questions will serve to future researchers as added benefit when designing their own measurement campaigns.

For this paper, we decided not to include any schematic diagrams in favor of keeping the length of the manuscript at a tolerable limit, but also in favor of not over-generalizing our results. After all, our analyses rely on only four days. As both the Reviewers have already stated, the representativity of our findings, namely of the three regimes, highly depends on the valley investigated. For instance, if a valley is not curved, or does not experience sufficiently strong daytime upvalley flows, it would not experience the third regime characterized by a CVV. On the other hand, if a valley is curved but semiarid, it might experience a much later $H$ sign reversal. Ultimately, the range of real valleys to which our findings can **directly** relate is extremely small. Nonetheless, the new method based on spectrum width detection of turbulent layers can find much broader usage.

- **Why does a diagnostic of the CBL height give insight on the general dynamics in the Inn Valley? Unfortunately, the diagnostic only seems to work before noon, when buoyancy is the dominant production process for TKE production.**

We have added an additional paragraph into Section 4.2 of the 2nd version on line 508, where this advantage, as well as the drawback the Reviewer rightfully brought up, have been elaborated in more detail.

- **What is the essential take-away when we can finally diagnose the CBL height not only in the vertical, but also in space?**

We interpreted the answering of this question as a good opportunity to help highlight the advantages of such a *quasi-Howmoeller*-style representation. Therefore, we added a new paragraph on line 605. Essentially, the spatial distribution provides the information on how representative a single vertical column is.

- **line 29: The substantial importance of horizontal shear was also shown by Goger et al (2018).**

Both of their related 2018 and 2019 publications have been added to the 2nd version on line 30.

- **line 24: "will lead" [...] "leading" please reformulate**

This sentence has been reformulated on line 23.

- **lines 30-48: This paragraph can easily be shortened towards "MoBLs are complex due to their multi-scale flow strucutre"**

Although we agree that the general theme of that paragraph fits the succinct description that the Reviewer suggested, we decided not to alter its somewhat lengthy structure as it is one of the pivotal literature review paragraphs. Furthermore, now with more space in the 2nd version of the main text, we did not find the need to shorten this particular paragraph.

- **lines 65-76: I am not sure whether the introduction needs this lengthy description on the disadvantages on ceilometers**

We agree with the Reviewer and have therefore removed the last two sentences of the relevant paragraph, starting from line 71.

- **Section 3 (or even before): Add a brief summary of the IOPs you are using for this manuscript (parhaps with an overview table of the most relevant information). I know that the IOPs are described extensively in previous publications, but a summary is necessary for the interested reader here to udnestand the rest of the manuscript.**

- **Here would be the opportunity to briefly discuss the differences between the single IOPs. If they are all similar, describe a typical diurnal cycle in the Inn valley (perhaps with a concept graph?) to prepare the reader on what to expect in the next chapters.**

We have introduced a new section immediately after the Introduction, titled *Typical evolution of the MoBL in the investigation area during August.* The goal of this section is to broadly describe the usual pattern of MoBL evolution in the investigation area, based on specific CROSSINN-based publications, but also some older ones. As a result, we have moved the brief introduction of the CVV, previously located near the end of the Introduction, to this new section, since it fits the storyline even better. We did not introduce a new table to accomplish this, but rather, a summary

of the most relevant features has been added (with appropriate citations to our past CROSSINN publications where interested readers can find more).

- **line 345: Which IOPs experience which large-scale forcings? Please elaborate.**

We combined this comment from the Reviewer with their previous comment above (regarding differences). We expanded the end of the former Section 3 titled *Spatio-temporal development of MoBL behaviour in the investigation area* with the relevant information regarding the differences in the IOP's large-scale forcing (or lack thereof) starting on line 387.

- **line 355: "the prevalence of downvalley flows at night": Looking at Fig. 7b, only IOP2b shows persistent down-valley flows during the night. Up-valley flows during the night are not typical for days, when the diurnal valley wind circulation dominates in the Inn Valley (e.g, Goger et al 2018, Lehner et al, 2019).**

- **Follow-up, what are the reasons that there are no down-valley flows in all the other (chosen) IOPs?**

We remind the Reviewer that the emphasis on IOP 2b, regarding only it having downvalley flow during nighttime, was already located in the sentence that follows the one cited by the Reviewer.

We argue that the primary reason for the lack of a more regular/sustained downvalley flow on IOPs 2a, 3 and 4 is the presence of low-to-mid cloud cover during the night. Presence of cloud cover, especially at this lower elevations, has the potential to disrupt longwave radiative cooling losses into space, thereby disrupting the necessary establishment of sufficiently strong horizontal pressure gradients that drive the downvalley flow itself. Furthermore, weak and intermittent rainfall occurred during the first few hours of IOP 4 (until roughly 06 UTC). These facts and accompanying explanations/hypotheses have been added to the 2nd version on lines 380 and 387.

- **line 357: "synoptic foehn influence": This is the first occasion in the manuscript where you mention foehn winds at all. Please describe them and their potential impact on the Inn vally boundary layer at an earlier opportunity (e.g., beginning of section 3 where you could outline the diurnal cycle in the valley and potential synoptic influcenes). Furthermore, Plavcan's foehn diagnostic applies to the city of Innsbruck, 30km west of the I-Box area. How can you make sure that this diagnostic can be also apllied to the I-Box station? Did you check, e.g., the upstream slope stations Weerberg and Hochhaeuser?**

We thank the Reviewer for this comment. We have inquired about the relevance of the Plavcan diagnosis with Ivana Stiperski (who looked at the i-Box stations in general in her earlier work, not any of the CROSSINN periods), and based on her experience, the diagnostic (for Innsbruck)

is sufficiently representative, or rather, adaptable to Kolsass. At Hochhaeuser, foehn might occur more frequently and the diagnostic for the site Ellögen (quasi-mountaintop location) might indeed be more appropriate.

Furthermore, foehn trajectories and the mechanisms at Kolsass differ significantly from those in Innsbruck. Kolsass lacks a deep side valley with a low pass like in Innsbruck - the Weer Valley to the southeast of Kolsass (Fig. 1 in the main text) is not nearly as deep and wide as the Wipp Valley. What we have found in limited previous studies (e.g. GOhm et al, 2009) is that the foehn flow at lower levels at Kolsass/Schwaz is more of a westerly wind, i.e. the foehn current from the Wipp Valley which is deflected in downvalley direction by the Nordkette. In fact, the 2nd half of IOP 10, which was diagnosed as a strong foehn event, experienced intense southwesterly, "downvalley" flow during the occurrence of foehn at Innsbruck (not shown). Similar can be seen from the simulations performed by Gohm and Mayer (2004), Zängle (2009), and Umek et al (2021), their Fig. 6. It can also be the combined air stream from other side valleys. An additional aspect preventing the detection of foehn at Kolsass could be, at least during wintertime but not necessarily during CROSSINN, the persistence of the cold air pool blocking foehn penetration. We therefore assume that the total number of foehn hours at Kolsass is significantly lower than at Innsbruck. To summarize: for strong foehn, the Innsbruck foehn diagnostics may be a proxy for foehn (or at least its effect on the MoBL) at Kolsass, but not necessarily for weak foehn. The same is true for Hochhäuser/Ellbögen.

- **line 365: Previous studies from the Rivera Valley (Weigel and Rotach, 2004) suggest that sensible heat fluxes (H) from slopes might have a larger impact on the valley boundary layer structure than H from the valley floor. Why do the authors only elaborate on the valley floor H, while there are observations from the other i-Box stations (e.g., Terfens, Eggen, Hochhäuser)?**

We remind the Reviewer that we already do show $H$ from the two slope stations in the former Fig. 10c, currently Fig. 8c. We now realize that this might not have been most ideal, and have included additional statements starting from line 584 regarding the behavior and magnitude of those two stations with respect to Kolsass, when interpreting Fig. 10 (now Fig. 8 in the 2nd version). However, as we will show in response to one of the Reviewer's later comments, it will become obvious that $H$ at Eggen was on average the highest, while $H$ at Kolsass generally changed the sign in the afternoon first - with $H$ at Hochhaeuser generally in between these two. Overall, these findings are in line with Lehner et al (2021), who have studied the diurnal cycles of the components of the surface energy budget in much greater detail and across more of the i-Box sites than just the three of interest here. However we argue that drawing conclusions based on just a few i-Box stations, regarding slope influences (similar to the Rivera Valley), though potentially valid, would be incomplete without more spatially-focused measurements (e.g. scintilometer-based $H$ measurements) or numerical modelling with appropriately fine horizontal resolutions. Therefore, we do not expand the 2nd version with more visualizations of $H$ at the slope sites.

- **line 367: Early turn of sensible heat fluxes: Can this turn of sensible heat fluxes also be connected to advection processes by the valley winds, a similar processes as negative SH fluxes during strong foehn flow (Umek et al (2021), their Figure 4)? How do you argue the influence of local vegetation when this turn of H was already observed 35 years ago at a different location in the Inn Valley (Vergeiner and Dreiseitl (1987), their Figure 8) or from the Rivera Valley (Rotach et al, 2008, their Figure 5)?**

We thank the Reviewer for this comment. Indeed, several past studies have found such an early $H$ sign shift to take place in multiple valleys worldwide. Of specific interest for our study is that of Lehner et al (2021), who have focused on the same i-Box area of interest as we do. At the moment, the unusually early $H$ sign reversal is held to primarily be a result of low Bowen ratios on one hand (e.g. forested and cultivated valleys), and on the other hand to coincide with the reversal of along-valley flow from downvalley to upvalley. Both of these hold in our case as well. Unfortunately, with our own instrument setup, it was not possible to ascertain the exact impact horizontal advection and/or horizontal flux divergence terms may have on $H$ sign reversal.

- **line 375: "assuming H is the sole driver of the CBL": How sure can we be of that assumption in complex topography? On the one hand, the authors wrote a very lengthy introduiction about the complexity of mountain boundary layers, but in the end they use a column approach with H from a single station, although there are more observations available.**

During the review process we also found that this assumption is needlessly invoked, therefore we removed that part of the sentence from the revised manuscript on line 420.

Regarding the use of the column approach above Kolsass only: We agree that, in order to keep things as simple and interpretable as possible whilst acknowledging complexity of the Inn Valley, we relied on an interpretation model based on a 1-D column, via $H$, $\Gamma$, and $w_L$. Perhaps a similar analysis could have indeed been made for Eggen and Hochhauser, however we did not see any additional benefit compared to what has already been done for Kolsass. As we already emphasized in the Summary section of the 1st version, having cross-valley $H$ transects from scintillometers, or at the least regularly spaced surface flux towers along our transect, would be necessary to have these three variables sampled at comparable horizontal resolutions - which was not the case during CROSSINN.

- **line 400: Subsidence values are compared to idealized simulations - are there no observations from other campaigns available? What about MAP-Rivera (e.g., Weigel et al, 2006)?**

Subsidence values from either remote sensing or in-situ measurements over valleys are surprisingly

rarely reported. The already cited study by Adler and Kalthoff (2014) is one of those studies, where subsidence values of the order of magnitude similar to the one we observed is reported. The values reported by Weigel et al (2006), on the order of -0.3 to -1 m s$^{-1}$, refer mostly to the period with strong upvalley flow up the Rivera Valley, which they have shown to generate a CVV-type circulation due to the upvalley flow curving in an anticlockwise fashion. In our case, we are focusing on the valley-wide subsidence values **before** the onset of the CVV, i.e. during the growing CBL regime.

- **line 407: The influence of the up-valley wind leads to a stabilization of the Inn Valley boundary layer, visible in previous simulation studies and the CROSSINN radiosonde observations. The term "well-mixed" might not be a fitting choice here.**

We went with the choice of "well-mixed" to signify that we see that, at those late afternoon times, the valley atmosphere is still well-mixed (or highly turbulent), though we cannot attribute it to surface-driven buoyancy anymore (since $H$ became negative). We considered also the term "turbulent layer", but in the end, we went with "well-mixed".

- **line 412: "foehn-driven turbulence": What do you mean with this term? Considering the TKE budget equation, TKE can be generated by buoyancy and/or shear. You can check in the i-Box stations, whether the TKE measurements are similar between your convective and foehn days, and can also calculate bouyancy production and the vertical shear to check on the source of turbulence (at least at Kolsass and Terfens). Furthermore, could you determine TKE (and budget) values from your aircraft observations?**

In the revised 2nd version, we have replaced *foehn-induced* with *shear-induced* (with an honourable mention of foehn within added parentheses in the same sentence) on line 459, to emphasize the lack of surface-driven convective influence in large spectrum width values at those elevations.

Although the aircraft sampled measurements at 100 Hz which would likely be sufficient for TKE calculation, unfortunately aircraft did purposefully not fly during any of the *foehn-contaminated* episodes, since we focused our flight hours on **quiescent** MoBL development conditions only (IOPs 1, 2b, 4, 6, 7b), thus making any such comparison of foehn and non-foehn situations impossible.

- **line 415: "horizontal convergence of upslope flow branches detaching from the slopes": This is true, but slope flows are likely eroded by the up-valley wind after 12 UTC (Rotach et al, 2008, Goger et al, 2018, their Figs 5 and 7). Furthermore, how sure are you on the development of slope flows in this non-ideal boundary**

**layer in the valley with foehn influence?**

We agree that the upvalley flow eventually overpowers the slope flows on both sides, since they are most likely to prevail during *free convective* conditions, i.e. when the along-valley flow switches from downvalley to upvalley. For the plateau, the *land-locked* elevated spectrum width at the corner of the plateau, as well as timelapse animations (of coplanar-retrieved $w$ and spectrum width) reveal that the upslope flow + thermal coupling there does indeed take place. As for the southern sidewall, timelapse animations (of coplanar-retrieved $w$ and spectrum width) once again reveal presence of upslope flows, together with time-distance plots of radial velocity from the Windcube lidar which was deployed at Hochhaeuser. Since that lidar scanned below the horizon, it was able to partly detect the signal of an upslope component (negative values of radial velocity). In Figs. 6 (IOP 2a) and 7 (IOP 4), leftmost panels show the radial velocity measured at an elevation of -13 degrees below the horizon by the lidar at Hochhaeuser (these figures were made as a quicklook at the end of 2019, not long after IOP 10, so we apologize to the Reviewers for their crudeness). The time period of interest is from approximately 08:00 UTC ($H$ increases) until 13:00 UTC (onset of strong upvalley flow and eventually CVV). An along-slope component of upslope flow, manifested as negative radial velocity from the lidar to roughly 400 m distance from it, is evident in both Figs. 6 and 7 (circled regions).

[Figure]

**Fig. 6:** -13 deg elevation beam extracted from the WLS200s-115 lidar at Hochhaeuser, for IOP 2a.

[Figure]

**Fig. 7:** -13 deg elevation beam extracted from the WLS200s-115 lidar at Hochhaeuser, for IOP 4.

- **line 418: Instead of speculating, you could check your radiosonde measurements whether they give any information on the mountain-to-plain wind circulation, e.g. by a shift in wind direction above crest height?**

Given the moderate westerly flow at elevations between 2000 and 5000 meters AVF on all four IOPs, it is highly unlikely that a clear signal of a mountain-to-plain wind circulation (roughly northerly-to-northwesterly direction) could be teased out of the soundings. This does not however imply that no appreciable mountain-to-plain wind circulation occurred at all.

- **line 435: Now IOP3 is also under a strong foehn influence - Please write in the beginning of Sect. 3, which IOPs have considerate synoptic forcing. Then, the question could be raised, why the authors chose these IOPs for CBL height determination. For example, CROSSINN IOP10, has way less disturbance from synoptic flows.**

Section 3 now includes a brief explanation of the different types of synoptic forcings each of the IOPs was characterized by on line 387. As for avoiding including IOP 10 (which was not entirely quiescent either), we refer the Reviewer to one of their earlier comments above.

- **line 441: "Increase in specturm width": Just curious, how is this an indicator for turbulence?**

In its simplest form, the longitudinal spectrum of flow velocity typically assumes the form of a reverse U-shape, where one side falls off at a slope of -5/3 (inertial subrange), while the other rises at a slope of +1 (energy-production range). When such a spectrum curve is plotted as a function of frequency times variance (i.e. the spectrum itself), the area underneath the curve is directly related to turbulence energy, i.e. TKE. That area can be increased with the increase of the peak of the spectrum curve, which by itself also increases the width of the area underneath the curve. This width expansion is ultimately reflected in greater quantities of spectrum width analyzed here.

More colloquially speaking, we could also say that since spectrum width reflects the degree of turbulence, an **increase** of spectrum width reflects an **increase** of turbulence.

- **line 474: "up to 200 m deeper above the plateau than over the southern sidewall", Why? Is this due to differential heating? Is H at the South-facing sidewalls larger than at the North-facing slopes?**

That is correct. As highlighted a few comments above, the 2nd version does contain a bit more information regarding the generally higher $H$ found at Eggen.

- **line 485: "plateau-locked upslope flows": What do you mean and where do you see this?**

We refer to the elevated spectrum width region, present across all IOPs, during the convective regime at x=+1200 m in Fig. 9a-h (1st version of the manuscript). The thermals at the edge of the plateau do indeed *come and go* at varying stages of their development. However, their persistent signal in the 1-hr average across all IOPs corroborates that this feature is indeed tied to the terrain.

- **line 492: "low-level upvalley flow jet" ... you mean the jet of the up-valley flow? You can determine the jet maximum from your radiosonde observations?**

That is correct. We can determine it from radiosonde observations (not done in this paper). Even better perhaps, we have determined it in B21 based on VAD-derived wind speed and direction from a HALO Photonics Streamline lidar that was deployed at Kolsass for all IOPs - please see Fig. 5a above.

- **line 509: "We will focus only on IOP 2a." Why? It would be an excellent improvement for the manuscript if you would show all four IOPs and then discuss the differences again. This would also highlight the different regimes observed.**

In the 1st version, we decided not to include the other three figures due to space limitations. Now that those space limitations have been avoided, we could certainly include them. However, we still decided not to do so, as they *tell* essentially the same story as the IOP 2a figure - just with more data gaps ... For completeness, we have included the remaining three figures here (Figs. 8-10).

[Figure]

**Fig. 8:** Caption is the same as in the manuscript.

[Figure]

**Fig. 9:** Caption is the same as in the manuscript.

[Figure]

**Fig. 10:** Caption is the same as in the manuscript.

- **lines 535-539: This was already done by Weigel and Rotach (2004). It would be a valueable insight whether this method of applying a different H also works for the Inn Valley.**

We agree with the Reviewer that it might be extendable to the Inn Valley, however we do not consider it as relevant to the present analysis and discussion.

- **line 532: "given the CVV influence": Is this really just the CVV - or just the up-valley flow?**

In a straight valley, an intense upvalley flow would not produce any secondary circulations that would generate significant upward/downward motions. In our slightly curved section of the Inn Valley, that however is the case, as evidenced by the repeatability of the CVV whenever upvalley flow strenghtens sufficiently (B21).

**Figures**

- **All figures: The front size in the figures varies a lot. Please be consistent.**

We are aware of this caveat, and acknowledge the fact that the font size varies from figure to figure. Ideally, if we only had figures with simple layouts, e.g. 1x1 or even 1x2 subplots, we would be able to stick to a constant axes-to-fontsize ratios. Unfortunately, that is not the case here, since we have figures ranging from a 1x1 layout all the way to 3x4 and even 5x3 subplot layouts. At the expense of having a fixed axes-to-fontsize ratios, we would then sacrifice legibility, since the proportionately small font size would be barely visible in these latter extremes.

- **Figure 1: You could add somewhere in a small box the general location in the Alps (or Europe).**

A small inset map has been introduced to Fig. 1 in the upper left corner. Its caption has been updated accordingly.

- **Figure 2 (and all future cross-sections in that style): Add South (S) and North (N) on the sides of your figure so that orientation is easier.**

We have added S and N only to the first figure (Fig. 2 in the new version). In any figure that follows, we only expanded its respective caption with e.g. *Orientation of the cross-valley transect is the same as in Fig. 2.*

- **Figure 3: Maybe I've missed it, but what unit is a.u.?**

Arbitrary units - please see earlier related comments.

- **Figure 4: Please add the day of your IOPs, otherwise the reader can not follow your seasonality argument.**

Dates have been added to what was previous Fig. 4, now Fig. S5.

- **Fig 7d: What's going on with $z_i$ from IOP2a? Please adjust the range of your figure.**

The y-axis range in this figure does not exceed 1100 m AVF, since this threshold has previously been used to remove potentially erroneously detected $z_i$ with the bottom-up exceedance threshold method. It can be seen from Fig. 8a (1st version of the manuscript) that, as the CVV strenghtens after 13 UTC, no cyan markers are present above 1100 m AVF.

- **Fig8: Where do you describe the CBL evolution of Fig8c,d? Is this not worth mentioning?**

Both of the lower subplots of former Fig. 8 were already described in their respective section, however we realize now that the detail dedicated to them was perhaps insufficient. We have expanded this section with a bit more detail, also citing these two subplots more directly in the text on lines 488 and 495.

---

## Author Response (AR2)

**Response to the Reviewers' comments on the manuscript**
**Exploring the daytime boundary layer evolution based on Doppler spectrum width from multiple coplanar wind lidars during CROSSINN* [egusphere-2023-1977]**
**ROUND 2**

Babić et al

We thank the anonymous Reviewer for their constructive comments, critiques and suggestions, which have helped us further advance the quality of this manuscript. Individual reviewer comments below are in bold font, while our responses are in regular font. Please note that, following the Copernicus guidelines for the write-up of final responses, we applied the *latexdiff* tool to the 2nd and 3rd versions of the manuscript. The resulting PDF has been attached with this document. The lines we refer to throughout this response correspond to lines in this *latexdiff*-generated document.

**I thank the authors for their detailed answers to my questions and I appreciate the first author's dedication to the manuscript despite leaving academia. However, the author's current professional situation should not alter the manuscript's quality, and furthermore, there are four more co-authors on the list who could also contribute to the improvement of the manuscript.**

We thank the Reviewer for these remarks and for giving us an additional opportunity to further enhance the manuscript's storyline.

**The revised manuscript was shortened (by moving most of the methods section into a supplement) and a new chapter on MoBL evolution the valley was added (this is much appreciated), but asides from these efforts, not much changed in the revised version, despite multiple questions and concerns by both referees. By this, I mean that many questions and concerns from me (and also from the other referee) were answered in a detailed way in the response to referees document (thank you!), but were not included in the manuscript. Since the authors already made the effort in answering and creating additional figures, why do they not include the answers in some way in the manuscript? Please be aware that referee's questions are always also potential questions raised by future readers of the manuscript.**

We agree with the Reviewer regarding the reader, at one point, posing the same and/or similar questions. Ultimately, when finishing up the revised version of the manuscript during the first round, we decided to err on the side of caution, given that both the reviewers raised very valid

concerns regarding the manuscript's initial excessive length.

**I would really encourage the authors to include a concise discussion section to put their research into broader context, because right now, this study is still very localized and limited to a few hours (!) of few days (three, if counted correctly) within a small section of an Alpine valley (peak-to-peak distance of 10km). I would like to remind the authors on WCD's guidelines on submitting research articles (`https://www.weather-climate-dynamics.net/about/manuscript_types.html`): "Research articles report substantial new results and conclusions from scientific investigations of dynamical processes in the atmosphere within the scope of the journal. Please note that the journal scope is focused on studies with general implications for atmospheric science rather than investigations which are primarily of local interest." Therefore, I would still suggest a second round of major revisions to give the authors the chance to improve the manuscript and add important discussion points to the manuscript.**

Once again we thank the Reviewer for granting us a chance to better our manuscript even more. As per the Reviewer's suggestion, we have introduced a Discussion section into the revised manuscript to put our research into a broader context, given the scale disparity and generalization of our results to other valleys worldwide. We would like to point out that, by such a line of reasoning, **every** study focusing on complex mountainous terrain (valleys being the most studied setting within such terrain) suffers from limited generality. Indeed, there are no two identical valleys in the world, at least in the sense of land use characteristics, horizontal and vertical dimensions, sidewall angle, etc. This problem is well known in the mountain meteorological community. However, we do not necessarily see this as a critically hindering drawback prohibiting peer review publication, but rather as each such study contributing a *data point* to a hypothetical parameter space bounded by the variables listed in the previous sentence. Our study contributes with an additional point to this parameter space. With each new measurement campaign (e.g. TEAMx Observational Campaign being the next major one, `http://www.teamx-programme.org/observational-campaign/`), this parameter space is continually being populated with more and more data points, which ultimately advances our knowledge of mountain meteorological processes.

The new Discussion section has been added starting on line 550 (for brevity of this document we decided not to copy/paste all the text here).

**I can understand that not all of referee's remarks are equally useful, but I would like to mention examples from the response to reviewers document which would improve the manuscript:**

**\*) page 6, R1's inquiry about the choice of the CBL threshold of 1100m. Wouldn't it make sense to add the reasoning behind it in the manuscript discussion?**

We have added the response to Reviewer R1's question regarding this threshold into the Supplement starting from line 162 (via Fig. S6), rather than into the discussion in the main text, since the threshold in question is defined in the Supplement.

**\*) page 10, R1's question on the presence of a LLJ;**

We have added the response to Reviewer R1's question regarding this LLJ/high spectrum width near the surface into the main text on line 420:

*Extremely high values of spectrum width are related to significant turbulence shear production at this time of day, which results from large vertical wind shear $dU/dz$ in the layer below the jet maximum, and possibly also from significant momentum covariance $\overline{u'w'}$ (Stull, 1988).*

**\*) page 16, R2: Does the new CBL height determination method help us [...]? - The author's answer to this question would perfectly fit in a discussion section.**

This question by the Reviewer from the first round of revisions has now been addressed via the newly added Discussion section.

**\*) page 19, R2: Rivera Valley: Thank you for this detailed answer - the manuscript would benefit if you added this to the manuscript. I don't think that a direct quantitative comparison is necessary - a qualitative one is useful as well, especially these sentences: "However we argue that drawing conclusions based on just a few i-Box stations, regarding slope influences (similar to the Rivera Valley), though potentially valid, would be incomplete without more spatially-focused measurements (e.g. scintilometer-based H measurements) or numerical modelling with appropriately fine horizontal resolutions."**

We thank the Reviewer for this suggestion. With the exception of the numerical modelling remark (which is already covered at various parts of the Conclusion section), the proposition Reviewer suggests is already (though not word-for-word) present in the last three sentences of the Conclusion section found on line 648:

*On the other hand, although we have been able to provide highly resolved cross-valley transects of $z_i$ and $w_L$, our conclusions were nonetheless restricted to having just three discrete sites offering $H$. To truly explore the CBL growth framework, similar highly resolved transects of $H$ are necessary, a demand potentially fulfilled with an array of carefully sited scintilometers (Ward et al, 2017). We are confident that the upcoming Multi-scale Transport and Exchange Processes in the Atmosphere over Mountains (TEAMx) programme and experiment (Serafin et al, 2020; Rotach et al, 2022) will offer the necessary means and resources to address these remaining challenges.*

**\*) R2: "well-mixed" ABL: I am sorry to say this, but it is problematic to call a bo-**

undary layer with a negative heat flux "well-mixed", because it is usually associated with CBLs or forced convection. (AMS Glossary: "The terms mixed layer, convective mixed layer, and convective boundary layer commonly imply only the buoyantly stirred layer." `https://glossary.ametsoc.org/wiki/Mixed_layer`). At 16UTC, buoyancy is likely negative in the Inn Valley (cf. Goger et al, 2018, their Figs 5 and 7). I would strongly suggest either "turbulent layer" or "shear-driven".

Following the Reviewer's suggestion, we replaced all *well-mixed* labels with *turbulent* throughout the revised manuscript.

**\*) page 24, line 509: Thank you for these interesting figures! I have to disagree with the authors, I think adding all the IOPs would benefit the manuscript - especially because it shows that the three IOPs exhibit similar conditions, which might allow for some more general conclusions? I would like to remind the authors that the extreme localization and restriction a few days is still a weakness of the manuscript.**

Following the Reviewer's recommendation, we have moved the three figures in question from the Supplement into the main text, while also adding specific descriptions for the conditions observed for each of the three IOPs (IOP 2b, IOP 3, IOP 4) into a new paragraph. The initial description for IOP 2a is still the main one, and we envisioned the new additions to simply describe the differences found on those other three IOPs compared to IOP 2a. To summarize, the new paragraph has been added starting from line 533:

*Concerning the behavior of the other IOPs, the overall similarity between IOPs 2a and 4 is evident, i.e. somewhat higher $z_i$ values over the valley floor and the southern slopes between 12:00 and 14:00 UTC (Figs. 8a, 11a). As already mentioned in Sec. 4.1, the deeper CBL on these two IOPs goes along with less resistance during their deepening, as their $\Gamma$ values decreased in time from roughly 4 to 1 K km$^{-1}$ (Fig. 8c, 11c). The pronounced cross-valley asymmetry in the w field on IOPs 2a and 4 due to the CVV after 14:00 UTC, although similar, differs on IOP 4 owing to relatively lower CBL still detectable by the bottom-up method. Furthermore, during the CVV phase (Fig. 11a), i.e. during the third regime (Fig. 7l), the deeper detected CVV over the plateau compared to the southern part of the valley is another significant difference compared to IOP 2a. Figure 9a indicates that IOP 2b had evidently the shallowest CBL of all four IOPs. This behavior holds for the whole N-S cross section, though $z_i$ reached somewhat higher over the southern slope and plateau than over the valley floor until 14:00 UTC. As mentioned before, on this day a CVV did not develop in the afternoon (indicated by a nearly symmetric w field, Fig. 9b) while $\Gamma$ was quite high in the morning (Fig. 9c). On IOP 3, despite the developing foehn, the CBL was nonetheless able to develop in a manner similar to the other IOPs until about 11:00 UTC (Fig. 10a). Afterwards, the continually descending foehn layer (Fig. 6g,k) reached the up to then undisturbed CBL, resulting in the inability of the bottom-up method to yield meaningful $z_i$ anymore.*

**Figure edits**

- As per the request by the editorial support staff, we had to redo the color schemes on Figs. 1, 5 and 6 to conform to readers with color vision impairments. Therefore these figures have been modified in the revised version (just the visuals, the content remained the same as in the previous version of the manuscript).